# Deactivation and Regeneration of Zeolite Catalysts Used in Pyrolysis of Plastic Wastes—A Process and Analytical Review

**Vivien Daligaux \*, Romain Richard \***  **and Marie-Hélène Manero \***

Laboratoire de Génie Chimique, Université de Toulouse, CNRS, INPT, UPS, F-31030 Toulouse, France
* Correspondence: vivien.daligaux@toulouse-inp.fr (V.D.); romain.richard@iut-tlse3.fr (R.R.);
marie-helene.manero@iut-tlse3.fr (M.-H.M.)

**Abstract:** In catalytic industrial processes, coke deposition remains a major drawback for solid catalysts use as it causes catalyst deactivation. Extensive study of this phenomenon over the last decades has provided a better understanding of coke behavior in a great number of processes. Among them, catalytic pyrolysis of plastics, which has been identified as a promising process for waste revalorization, is given particular attention in this paper. Combined economic and environmental concerns rose the necessity to restore catalytic activity by recovering deactivated catalysts. Consequently, various regeneration processes have been investigated over the years and development of an efficient and sustainable process remains an industrial challenge. Coke removal can be achieved via several chemical processes, such as oxidation, gasification, and hydrogenation. This review focuses on oxidative treatments for catalyst regeneration, covering the current progress of oxidation treatments and presenting advantages and drawbacks for each method. Molecular oxidation with oxygen and ozone, as well as advanced oxidation processes with the formation of OH radicals, are detailed to provide a deep understanding of the mechanisms and kinetics involved (direct and indirect oxidation, reaction rates and selectivity, diffusion, and mass transfer). Finally, this paper summarizes all relevant analytical techniques that can be used to characterize deactivated and regenerated solid catalysts: XRD, $N_2$ adsorption-desorption, SEM, $NH_3$-TPD, elemental analysis, IR. Analytical techniques are classified according to the type of information they provide, such as structural characteristics, elemental composition, or chemical properties. In function of the investigated property, this overall tool is useful and easy-to-use to determine the adequate analysis.

**Keywords:** plastic waste; recycling; catalytic pyrolysis; regeneration; zeolite catalysts; coke

## 1. Introduction

During the last century, plastics have become an essential material for enhanced life quality and are nowadays present in our lives on a daily basis. Use of plastics has enabled innovations and technological progress in many domains such as construction, healthcare, electronics, automotive, packaging, and other specific applications [1]. Consequently, global plastic production has exponentially increased during the last 50 years to comply with the needs implied by population growth. While 15 Mt of plastics were manufactured in 1964, worldwide plastic production reached 350 Mt in 2018 and could double within the next 20 years [2]. The consequent rise of plastic consumption has led to plastic accumulation. Around 275 Mt of plastics waste is generated each year around the world [3]. Indeed, the technologies developed for plastic recycling are not in capacity of handling the integrality of the increasing amount of waste. Based on European statistics in 2016, 27.3% of the plastic waste is disposed to landfill, 31.1% is recycled, and 41.6% is used for energy recovery [4].

To face the environmental issues caused by intensive plastic use, the necessity to develop efficient and viable recycling methods has risen as a worldwide challenge. Mechanical recycling is currently the main process used for plastic waste reuse. Other techniques, such as chemical recycling, have been heavily investigated recently to provide alternative

recycling [5,6]. Among these chemical recycling processes, catalytic pyrolysis has been identified as a promising method for plastic waste revalorization, by converting polymers into basic chemicals used as feedstock [1,7,8]. Concerning the different catalysts used in pyrolysis and recycling processes, zeolites appear to be the most relevant catalysts used in the last decade [9–12].

However, similarly to many other catalytic processes, industrial application of plastic waste catalytic pyrolysis is hindered by fast deactivation of catalysts. The science of catalyst deactivation has been steadily developed and its literature has expanded over the years, including books [13,14], comprehensive reviews [15–17], as well as many papers and topical journal issues [18–22]. Most of them indicate that formation of coke is a very common deactivation pathway in industrial processes involving organic compounds in the presence of heterogeneous catalysts.

The following work reviews the different methods used to remove these carbonaceous deposits for catalyst regeneration, with a particular focus on zeolite catalysts used in pyrolysis of plastic waste. Three main techniques exist to regenerate coked catalysts: coke oxidation (with air/oxygen or other oxidants), gasification (with carbon dioxide or water vapor), or hydrogenation [23]. Particular attention is given to literature dealing with oxidative treatments, using oxygen or alternative oxidants in milder conditions, as they represent the main interesting methods for coked zeolite catalysts.

Finally, relevant analytical techniques, developed to characterize and analyze coke deposition on catalysts and/or inside the pores of catalysts in order to understand mechanisms and kinetics of coke formation, will be presented. While techniques such as XRD (X-ray Diffraction) and TPO (Temperature-Programed Oxidation) have been and remain widely employed, SAXS (Small Angle X-ray Scattering) appears as an innovative tool with new analytical possibilities.

## 2. Pyrolysis Processes for Plastic Recycling

### 2.1. Plastic Waste and Recycling Methods

#### 2.1.1. Plastic Production and Generated Waste

Plastic use has grown extensively during the last decades and is nowadays present in many different sectors, from everyday life to technical applications. As shown in Figure 1, combined packaging (household, industrial, and commercial) is the main plastic consuming segment and accounts for almost 40% of worldwide consumption [5]. Globally, due to the plastic demand for each segment, the main produced polymers are identified to be polyethylene (PE), with different possible densities, polypropylene (PP), polystyrene (PS), polyethylene terephthalate (PET), and polyvinyl chloride (PVC) [24]. Nowadays, plastics are mainly produced from petroleum-based feedstock. To provide an alternative to petroplastics and to face limitation of fossil energy sources, use of biosourced material to produce biobased plastics rise as an interesting and sustainable option because of their biodegradability and renewability [25]. Bioplastics are currently marginal with only about 1% of the annual plastic production, but this market is expected to continuously grow within the next years [26]. Based on their different properties, such as rigidity, ductility, insulation capacity and others, polymers are used for many applications in various domains. For example, polyethylene is very commonly used for packaging purposes. Different densities of PE are possible according to the need: low-density PE (PE-LD) is used for plastic bags and wrapping foils while high-density PE (PE-HD) packaging applications are detergent bottles or oil containers. PP has a lower density than PE-HD but has higher rigidity, making it a rather light and resistant material. This polymer is therefore extensively used in plastic industry for diverse applications such as car bumpers or storage boxes. The cumulative consumption of the five most used types of polymers represents almost 75% of the total plastic use, while more than the half of it is due to PE and PP [27]. Representing the majority of plastics found in landfill, PE and PP are preferentially chosen in recycling R&D investigations.

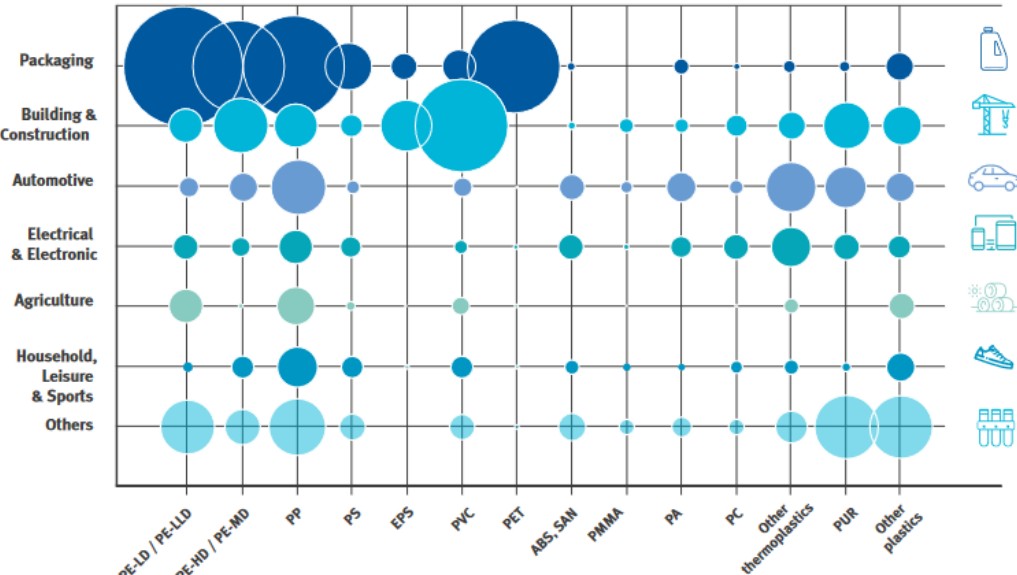

**Figure 1.** Plastic demand by domain and type of polymer in 2019. Illustration taken from report "Plastics—the Facts 2020". Source: PlasticsEurope [24]. Polyethylene—PE (LLD: linear low-density; MD: medium-density), polypropylene—PP, polystyrene—PS, polyvinyl chloride—PVC, polyethylene terephthalate—PET, acrylonitrile butadiene styrene—ABS, styrene acrylonitrile—SAN, poly(methyl methacrylate)—PMMA, polyamide—PA, polycarbonates—PC, polyurethane—PUR.

Even though plastic waste can originate from the previous year's production, estimation of plastic waste generated per year is usually approximated to the annual production, since almost 50% of the production is aimed for single-use applications and only 9% of the production comes from recycled plastics [28]. Despite recent efforts to limit waste generation and to improve recycling, an important remaining part of plastics is directly disposed to landfill or rejected in the environment, causing huge space occupation and dramatic environmental issues. Each year, between 8 and 14 Mt of plastics are dumped into the ocean causing irreversible damages to marine ecosystems and biodiversity [29]. Landfills also have a direct negative environmental impact by poisoning soils, altering land biodiversity, and by emitting greenhouse gases [30]. Because of their stable chemical nature, polymers are very persistent in the environment and may take more than 100 years to undergo natural degradation. Pollution due to plastic waste accumulation and spill in nature is therefore a major environmental issue and the combined reduction of waste and amelioration of recycling has become a worldwide problem. To achieve the model of a circular economy, aiming a complete reuse of plastic waste with the creation of a closed loop between production and waste management, many public and private international actors of research investigate new alternative methods to increase and to improve plastic recycling and revalorization [5,31].

2.1.2. Recycling Methods

The different recycling methods found in the literature can be split into different categories: mechanical, chemical, and biological recycling. Figure 2 represents the different available methods for plastic recycling and their implementation in the aimed circular plastic economy allowing theoretical endless reuse of plastic wastes. The mechanical pathway is already well known and widely applied at industrial scale and is currently the most commonly used method for plastic waste recycling. In fact, the term "recycling" is nowadays mostly associated with mechanical recycling since it represents 99% of the recycled quantities in Europe [32]. It consists in reusing and reforming plastic waste without changing its chemical structure to form other consumable products, such as clothes made from recycled bottles for example. This pathway involves different steps (collection,

sorting, washing, and grinding), which may be mixed, repeated several times or not applied according to the composition and origin of treated waste [5]. However, mechanical recycling shows limitations because of the restricted applications of obtained recycled products and the insufficient capacity facing the enormous quantities of global plastic wastes. Consequently, the need to develop alternative recycling methods have appeared and researchers have recently demonstrated a strong interest for chemical and biological pathways. These techniques have arisen from the recent intensive research for reducing the environmental impact of plastic waste. These alternative recycling processes go further back in the polymer production chain, by modifying the chemical structure of the molecule.

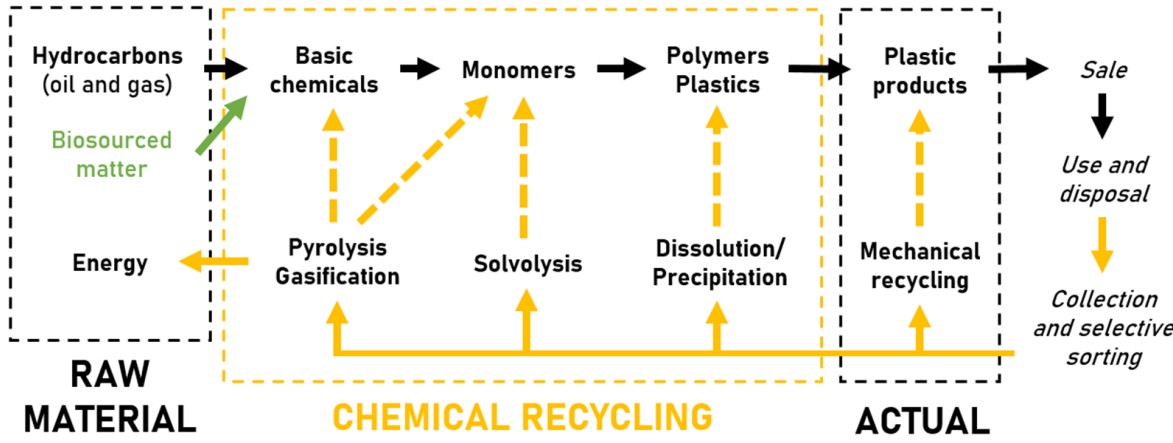

**Figure 2.** Schematic representation of the different existing methods for plastic waste recycling and their implementation in the circular economy.

While mechanical processes only use physical methods to sort and separate different types of plastics before grinding and reconditioning them, chemical and biological recycling directly affects the formulation of the plastic or the polymer with treatments modifying its chemical structure in order to obtain reusable raw materials [31]. Even though the term "recycling" is currently associated with the mechanical pathway due to its wide application, these treatments are also referred to as recycling methods since they ensure the recuperation, revalorization, and reuse of plastic waste. Chemical processes are based on the effect of solvents or temperature to transform polymer structure in order to obtain different products. Solvent-based methods have two possible outcomes: dissolution/precipitation of the polymer to obtain virgin-grade plastic with all additives removed, or depolymerization by solvolysis to recover monomers, offering greater liberty to produce another grade of polymer with different characteristics from the original one [33]. The first method is based on the solubility of a specific polymer in a particular or a combination of solvents. After a separation step where non-dissolved compounds are removed, an anti-solvent acts as a precipitating agent to recover the polymer in its solid and purified form. The main drawback of this purification is the difficulty to achieve complete removal of residual solvent that may affect polymer properties. In a similar way, solvolysis is based on the use of solvents to achieve removal of additives and to react, leading to the monomer furtherly polymerized again to form "new" plastic products. If purity is not sufficient for polymer synthesis, these recycled monomers can possibly be purified or mixed with conventionally obtained monomers. Different processes have been investigated for plastic monomer recovery and several processes follow this principle using different solvents: hydrolysis with water [34], alcoholysis using methanol (methanolysis [35]) or ethylene glycol (glycolysis [36]), along with phosphorolysis, ammonolysis and aminolysis [37–39]. Solvents are chosen in function of their affinity for the different polymers and their ability to cleave particular bonds. Indeed, only ester, ether and acid amine bonds can be broken

using solvolysis. Its application is consequently limited to polymers containing one of these groups (PET, PUR, PA, PC).

Biological recycling is also characterized by a modification of the chemical structure, occurring via an enzymatic degradation of polymers to form lighter molecules (monomer, dimer, olefins). Gamerith et al. investigated and proved the efficiency of enzymatic treatment to recover viable monomers for the production of polyesters from polymer blends containing mainly PET and PA [40]. Biodegradation occurs thanks to the action of microorganisms that, after a stage of adherence and colonization of the material, will break polymer chains and eventually form low-molecular weight products as well as byproducts, such as methane, $CO_2$ or water [6].

In addition to these methods using external reactants, other processes such as pyrolysis and gasification rely only on the effect of high-temperature treatments to degrade the polymer structure. The main difference between pyrolysis and gasification is the medium where plastics are heated: while pyrolysis is carried out in oxygen-free atmosphere, gasification medium contains a limited amount of oxygen. These processes enable the conversion of plastic waste to high-value liquid oils, solid char, and high-temperature gases. The yielded products are very similar to raw petroleum feedstock and can be used to be retransformed into polymers, but the obtained oils are generally revalorized as fuel due to their high energetic potential. Pyrolysis is however more embedded in a strategy of energy recovery and revalorization of wastes, transforming plastics into high-added value and reusable products. This promising process can be both thermal and catalytic. Pyrolysis has been heavily investigated during recent years: the mechanisms involved and the influence of operating parameters and reaction system over yielded products have been the subject of numerous articles and reviews [1,7,8,41].

### 2.2. Catalytic Pyrolysis of Plastic Waste

Catalytic pyrolysis of plastic waste has recently become a process of interest: its optimization as well as the understanding of its mechanisms and influencing parameters are actual research challenges. While thermal pyrolysis only relies on temperature for polymer cracking, the use of catalyst involves reactivity with an active surface that influences both polymer degradation and reforming reactions. Catalytic pyrolysis, especially with zeolite materials, offers a better selectivity and yields more high-value products than thermal pyrolysis, which contain impurities and residues. Moreover, on top of the improvement of products quality, the addition of catalysts in the pyrolysis process leads to the reduction of reaction temperature and retention time. It is widely accepted that catalytic acid sites favor cracking reactions. Consequently, catalytic pyrolysis yields lighter products compared to thermal pyrolysis, resulting in an increased gaseous fraction and reduced liquid fraction [42]. However, this quantity loss of liquid product is compensated by the rise of its quality, containing more molecules of industrial interest, such as light olefins or products having similar properties to automotive fuel such as diesel or gasoline. Figure 3 represents the comparative composition of pyrolysis yields for thermal and catalytic process of HDPE showing the difference in phase repartition and liquid composition [8]. This observation is explained by the enhanced conversion of heavy and long chain olefins to lighter compounds thanks to the reactivity of acid sites on the catalytic surface. Catalysis for pyrolysis of plastics can be homogeneous or heterogeneous. The latter is preferred due to the convenience for separation of catalyst from fluid product or remaining solidified molten polymer, while further separation process steps are required for homogeneous catalyst recovery after reaction. Either the solid catalyst can be directly mixed with the feedstock in the reactor, or it can be placed in a separate column where only the organic pyrolysis vapors can pass through. Direct contact strongly improves the cracking process, while in a two-stage reactor the catalyst only takes part in the following reforming reactions. Therefore, catalytic pyrolysis with direct contact yields better quality of liquid oils but is also more exposed to deactivation by coke formation or poisoning due to the deposition of other impurities, such as chlorine during PVC pyrolysis.

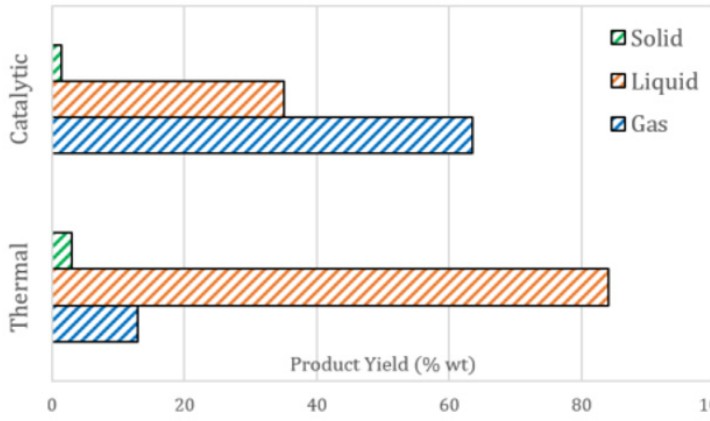

**Figure 3.** Product yield comparison between thermal and catalytic pyrolysis of HDPE. Graph adapted from Seo et al. data [43].

| Yield (%wt) | Thermal | Catalytic |
|---|---|---|
| Gas | 13 | 63.5 |
| Liquid | 84 | 35 |
| $C_6$-$C_{12}$ | 56.55 | 99.92 |
| $C_{13}$-$C_{23}$ | 37.79 | 0.08 |
| >$C_{23}$ | 5.66 | 0.0 |
| Solid | 3.0 | 1.5 |

Catalysts used for pyrolysis are part of three main categories: FCC (Fluid Catalytic Cracking) catalysts, zeolites, and silica-alumina catalysts. Their reactivity is due to metallic or acid sites contained over the surface area. FCC catalysts, heavily used in the petrochemical industry, mainly yield liquid oil as investigated by Lee et al. who reported between 80 and 90% of liquid oil production during pyrolysis of different polymer feedstock [44]. Silica-alumina and zeolite catalysts show different results with a more important gaseous fraction. This behavior is attributed to their improved acidic properties, depending on the Si/Al molar ratio. The increased cracking reaction rate due to acidity is responsible for the formation of lighter molecules, favoring gas formation. Sakata et al. compared liquid yield with catalysts of varying acidity and confirmed this statement: while a low acidity catalyst produced 74.3 wt% of liquid oil, high acidity ZSM-5 yielded only 49.8 wt% [45]. Among catalyst characteristics, structure and pore distribution have an influence due to the shape-selectivity of the reaction with different size distribution of products and intermediates. Many studies focused on the determination of catalytic pyrolysis yield with various feedstock, catalysts and operating conditions. In the literature, the most commonly used catalysts for catalytic pyrolysis are zeolites.

### 2.2.1. Zeolite Materials

Zeolite catalysts are aluminosilicate sieves with pores and channels forming a three-dimensional microporous framework. While silica-alumina catalysts have an amorphous structure, zeolites have a crystalline structure, composed of primary structural units T-O4 tetrahedron where T is the central atom, typically Si or Al, surrounded by O atoms connecting one unit to the other. Interconnection between those basic building units (BBU) gives rise to different possibilities of three-dimensional microporous structures with different geometries with specific structural properties [46–48]. Each zeolite is defined by its singularities: pore network with related porosity and tortuosity; channels and intersections in one, two or three dimensions; systems of cages connected by windows [47]. Applications of zeolites are usually dependent of their structure due to their shape-selectivity [49]. Zeolite reactivity is due to the presence of surface acid sites, which number and strength are determined by its composition (ratio Si/Al). Zeolites are widely used in heterogeneous catalysis for many catalytic applications in industrial processes. Among them, catalytic pyrolysis of plastic waste has been investigated with different zeolite types.

These different zeolite types can be HZSM-5, HUSY, Hβ and HMOR among others, as well as some natural zeolites. The most commonly investigated zeolite is HZSM-5 due to its higher catalytic activity, better selectivity and limited deactivation. Indeed, Garfoth et al. observed promising results for HZSM-5 catalysts compared to HUSY and HMOR zeolites during HDPE pyrolysis [50]. Within a same zeolite type, catalyst acidity can differ, as was investigated for different HZSM-5 catalysts with various Si/Al ratios [9]. Most of the

studies carried out with zeolite catalysts mainly focus on comparative studies with various catalysts and operating conditions to understand their influence over the distribution and nature of yielded products, but few comprehensive works deal with involved cracking mechanisms and deactivation reactions [10–12].

### 2.2.2. Pyrolysis Reaction Mechanisms

During catalytic degradation of plastics, as temperature rises, the polymer firstly melts and is dispersed over the catalyst surface where it is broken due to its reactivity with acid sites. Different mechanisms involving ionic and free radicals have been proposed by many researchers and have been summarized in comprehensive reviews [51]. Mechanisms of catalytic pyrolysis usually involve chain scission, isomerization, oligomerization, H-transfer, and aromatization. The initial step of polymer cracking is agreed to be the adsorption of the reactant molecule on the acid site where it is protonated to obtain carbonium ions. This intermediate is known to promote the cracking of molecules. The reaction rate is therefore mainly influenced by acid site strength, density, and distribution. Indeed, acid active sites of the catalyst support the cracking of olefinic compounds and favor hydrogen transfer reactions [52]. Catalytic cracking can proceed by end-chain scission when catalyst acidity is strong, forming olefins, or by random scission in weak acidic medium leading to the formation of waxes. Those primary formed products undergo further reactions to eventually produce low molecular weight compounds, as presented in Figure 4 [53]. It has been suggested that initiating decomposition reactions can only occur over external catalyst surface due to the important size of polymeric molecules. Further transformations take place at the internal surface as initial cracking products have lower molecular size and consequently can diffuse through the molten polymer and enter catalyst pores to undergo the aforementioned secondary reactions. The equilibrium reached with these reactions, sometimes competing with one another, yields light hydrocarbon molecules in different phases forming char, liquid, and gases, in which repartition and nature depend on the operating parameters.

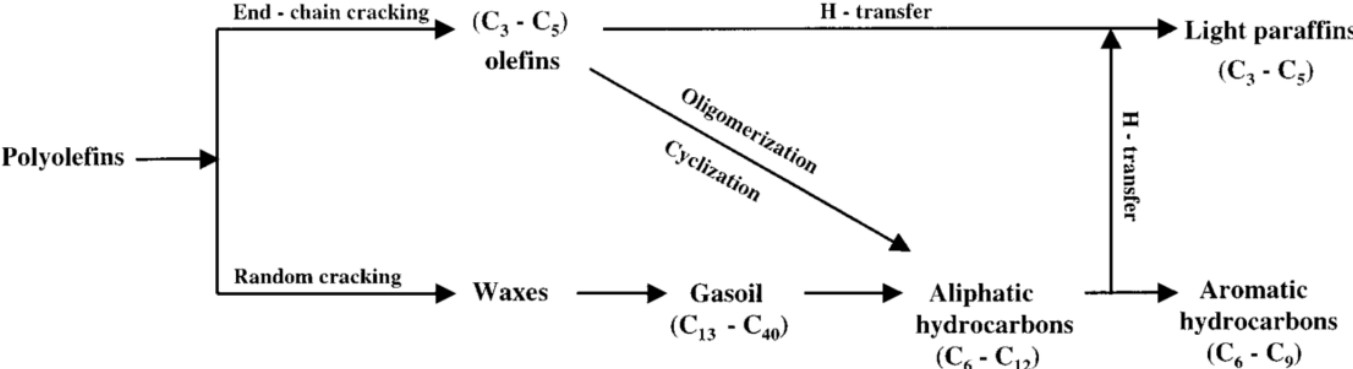

**Figure 4.** Reaction pathways for catalytic pyrolysis of polyolefins. Aguado et al. [53].

### 2.2.3. Influencing Factors

- **Operating parameters**

Among the different operating parameters, temperature is considered as the most important factor by influencing both repartition and nature of yield [54,55]. High temperature pyrolysis enhances cracking reactions, consequently favoring the formation of small molecules, whereas long chain hydrocarbons are produced at lower temperatures. During pyrolysis of plastic wastes in a semi-batch reactor, Lopez et al. found a variation of yield repartition between liquid and gaseous phase due to temperature [55]. From the comparison between yield at 500 and 600 °C, decrease of liquid phase quantity from 65.2 to 42.9 wt% is correlated with an increase of light gaseous phase from 34.0 to 56.2 wt%.

Temperature rise also enables the activation of secondary reactions leading to the formation of aromatics [56]. Consequently, a high gaseous product fraction is obtained at 600 °C while the liquid fraction is more important for lower temperatures between 300 °C and 500 °C, and the liquid fraction contains more aromatic products at 500 °C. The composition of each of these fractions is highly influenced by pyrolysis temperature. However, different heating ramps to reach the targeted temperature were tested during pyrolysis experimentations and did not appear as a major impacting factor. Reaction temperature has to be set in function of feedstock, as degradation temperature is different according to the polymer, as shown in Figure 5, which represents ThermoGravimetric Analysis (TGA) for different polymers. PVC is partially degraded at 300 °C and PS is completely decomposed at around 410 °C, while PET, PP, and PE degradation occur between 450 and 500 °C. As reactivity of plastics increase with temperature, the sole limitation is set by the maximum temperature before thermal damages of the catalyst. Products yield phase repartition varies slightly with the nature of the feedstock, as investigated in some studies carrying out pyrolysis experiments with different polymers or with mixed plastics [57].

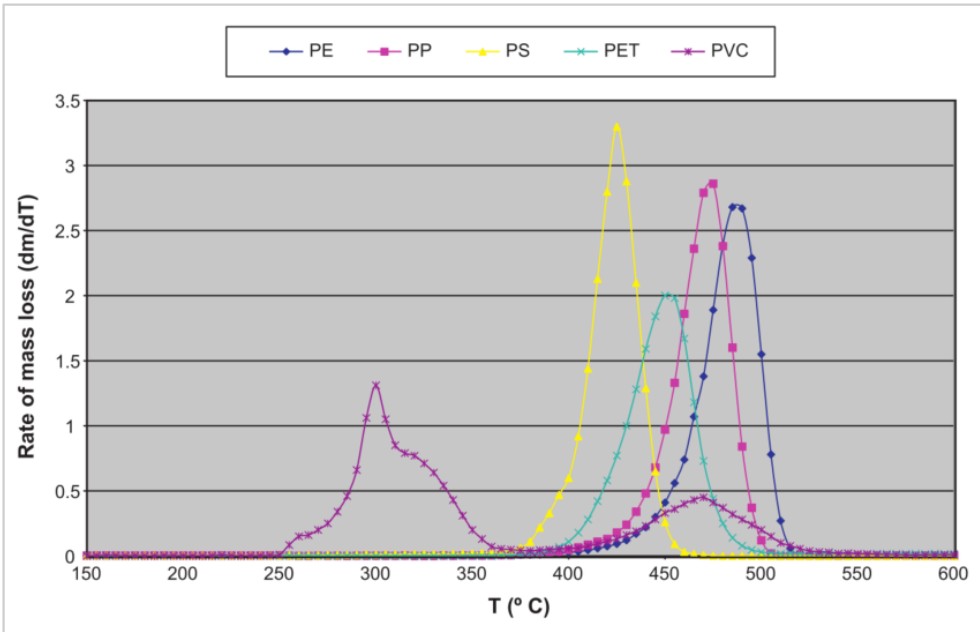

**Figure 5.** Thermogravimetric analysis (TGA) for determination of degradation temperature for different polymers. Lopez et al. [55].

Other operating parameters, such as pressure and retention time, were studied to determine their influence over pyrolysis yield. Murata et al. carried out thermal pyrolysis of HDPE at different pressures within the range 0.1–0.8 MPa in a continuous stirred tank reactor and observed an increase of gaseous products fraction from 6 to 13 wt% at 410 °C [58]. This effect tends to decrease as temperature rises. Moreover, pressure has been shown to affect the nature of products by shifting the average molecular weight to lighter compounds due to its direct impact over scission of C−C links. Pressure is consequently considered as an influencing parameter for pyrolysis and could be used to the control distribution of pyrolysis products, especially at low temperatures. However, the influence of pressure during catalytic pyrolysis still has to be investigated. Residence time, defined as the average time of retention of species in the reactor, may also influence product distribution since it directly affects conversion of reactants and secondary products to light hydrocarbons and non-condensable gases. However, Lopez et al. observed that product distribution did not change much between 30 min and 120 min pyrolysis experiments and determined that residence time is a highly influential parameter only up to 15 min of reaction, which is not sufficient for total reactant conversion [55]. Lee and Shen focused on

the composition of pyrolysis oil for different lapse time of reaction, between 0 and 400 min, and observed a varying repartition of known paraffin, olefin, naphthene and aromatic products between 350 °C and 400 °C [59]. Pressure and residence time can therefore be considered as influencing parameters for pyrolysis reactions, but remain temperature-dependent, their effect being less apparent at higher temperatures due to the temperature limitation in the process [60]. Based on the different studies available in literature, pyrolysis experiments are usually carried out at atmospheric pressure with a temperature around 500 °C during 30 min. The operating conditions will also vary depending on the nature of feedstock and the type of reactor used during the experiment.

- **Reactor type**

The type of reactor has an important impact during catalytic pyrolysis as it influences mixing of reactants and catalysts. Polymers are often used as provided in solid pellets of about 3 mm diameter, but can also undergo a grinding step to form solid powder (<1 mm). Ratio between catalyst and polymer commonly vary from 5 to 20 wt% of catalyst according to the study [12]. The type of reactor also affects residence time and heat transfer.

Use of batch or semi-batch reactors is very common in lab-scale experiments due to their ability to control operating parameters. In batch reactors, reactants are left during all reaction time, while a product extraction is performed with semi-batch reactors. They are particularly suitable for thermal pyrolysis. The main drawback is the variability of results due to the non-homogeneity of reaction medium. Seo et al. therefore embedded a stirrer in the experimental reactor for HDPE pyrolysis at 450 °C [43]. The provided agitation led to an increase of liquid fraction for both thermal and catalytic pyrolysis compared to experiments carried out in similar conditions by Sakata et al. without agitation [61]. Indeed, stirring provides appropriate heat transfer leading to better efficiency and viability of the process [62]. The addition of catalysts in this type of reactor has proved to influence the phase repartition of products as expected from previous observations [61,62]. Direct contact between plastic and catalyst is preferable for enhanced reactivity and improved liquid yield, but it also favors coke formation over the catalytic surface leading to deactivation. Combined with the high operating cost, the tendency to fast deactivation is the reason why this type of reactor is not recommended for catalytic pyrolysis for large scale production and remains mostly used for lab-scale experiments.

Continuous flow reactors, especially fluidized bed reactors, are suggested to be the most efficient reactor shape for the industrial application of catalytic pyrolysis because of improved heat and mass transfer as well as reduced deactivation providing to the catalyst an improved lifetime. The different types of continuous flow reactors are presented in Figure 6. Fixed-bed reactor is the easiest geometry to design but the packed catalyst bed causes issues related to mass transfer in the reactor, which could lead to reactor plugging due to the heavy and sticky nature of molten plastics. Moreover, the available catalytic surface area in a fixed bed is limited and catalyst efficiency is reduced. In studies carried out with fixed bed reactors, catalysts and plastic feedstock are not directly mixed: polymer is placed over the catalyst or even in a separate porous recipient [63–65]. To avoid direct catalyst exposure to molten polymer, some studies preferred to separate the pyrolysis reactor in two stages, one for pyrolysis followed by a fixed-bed column for reforming reactions with only pyrolysis gases passing through [66,67]. Use of a fluidized bed reactor solves some of the issues of fixed-bed as it provides a good mixing, leading to a higher accessible surface area and an improved mass and heat transfer. Consequently, catalytic pyrolysis needs shorter residence time and yields less variable products. Due to the industrial interest for this type of reactor because of low operating costs, pyrolysis in fluidized-bed reactor has been heavily investigated over the last decade [68–70]. In addition to these "classic" reactors commonly used for other processes, a new reactor geometry for catalytic pyrolysis has been investigated. Elordi et al. carried out pyrolysis experiments in a conical spouted bed reactor (CSBR) that, similarly to fluidized-bed reactors, insures a good mixing of the catalyst with reactants yielding high quality products [71]. With this reactor, a wider range of solid particle size and density can be handled and, according to Olazar et al.,

it helps by reducing the attrition and low bed segregation compared to fluidized bed [72]. Even though pyrolysis reactors and processes are often designed to postpone catalyst deactivation as much as possible, loss of catalytic activity via different mechanisms of deactivation remains an important challenge for wider use of catalytic pyrolysis.

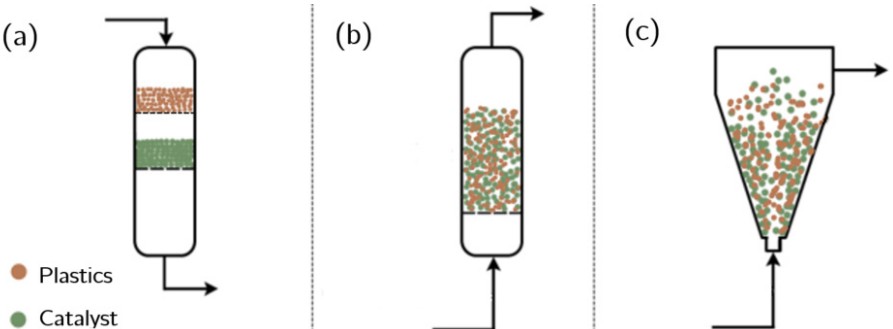

**Figure 6.** Different possible continuous reactor configurations for catalytic pyrolysis of plastic waste: (**a**) fixed bed reactor, (**b**) fluidized bed reactor, (**c**) conical spouted bed reactor. Adapted from scheme of Ochoa et al. [17].

### 2.3. Deactivation of Zeolite Catalysts

Catalyst deactivation is a well-known issue, as almost 80% of industrial processes involve catalysis. While catalyst deactivation is inevitable for most cases, some of its immediate and/or drastic consequences can be avoided, postponed, or even reversed. Therefore, deactivation phenomenon greatly impacts research, development, design and operation of commercial processes leading to a considerable motivation to understand and treat catalyst decay.

#### 2.3.1. Generalities

Industrial processes involving organic compounds in the presence of solid catalysts are widely carried out in industries, such as petrochemical industry, and catalyst deactivation is often observed. There are different paths for heterogeneous catalyst decay leading to catalyst deactivation and loss of catalytic activity. Generally, five main ways leading to catalyst deactivation are reported in literature: poisoning (1), gas/vapor-solid and solid-state reactions (2), mechanical failure of catalyst (3), thermal degradation and sintering (4) and fouling, coking, and carbon deposition (5). Poisoning is a very common deactivating mechanism and is known as a strong chemisorption of species that are not taking part in the reaction and are therefore "blocking" the active catalyst sites. This phenomenon happens when the chemisorption of these poisoning species is stronger than the affinity of reactants with active sites. Deactivation can also be due to unwanted reactivity of catalyst with species present in the reaction. Different types of reactions leading to deactivation can take place: gas-vapor solid reactions between the catalyst and gas phase turning active catalytic surface into inert phases or into volatile compounds leaving the reactor as by-products, or solid-state reaction with catalyst phase transformation during the process. Moreover, mechanical failure of the catalyst can occur in some cases and manifests in different ways: crushing, attrition, and/or erosion of catalyst pellets, all leading to a generally irreversible structural damage of the catalyst [16]. In this review, a particular attention is given to fouling/coking and thermal degradation as they represent the main deactivation risks for catalysts used for pyrolysis of plastics.

Thermal degradation of the catalyst is the loss of catalytic surface area resulting from the crystallite growth of catalytic phase or the loss of support/catalytic surface area due to support/pore collapse. These two phenomena are usually referred to as "sintering" in the literature. These processes generally take place during high-temperature reactions and are accelerated by the presence of water vapor. The principal sintering mechanisms, presented in Figure 7, are based on adatoms migration from small particles to larger ones (a), and

direct migration of small particles to agglomerate with larger particles (b). This process results in crystallite size growth, therefore reducing the active surface area and decreasing the catalytic activity. However, sintering is reversible and redispersion of metal particles is achievable to recover catalytic activity [73]. Nevertheless, structural degradation represents irreversible damages for catalyst crystallinity, leading to permanent loss of catalytic activity. As plastic pyrolysis is carried out at high temperatures, a particular attention has to be paid to the maximal temperature accepted by the catalyst to avoid deactivation by thermal degradation. Consequently, processes are designed to avoid this type of deactivation and thermal damages are usually not observed in normal operating conditions. Fouling and coking deactivation pathway is much more difficult to avoid as it usually occurs within operating conditions. The latter are defined by the deposition of chemical species, especially carbonaceous compounds referred to as coke, onto the catalyst internal and external surface, resulting in catalytic activity loss due to lowered access to active sites [74]. Being the main cause of catalytic deactivation in many processes, this type of deactivation has been heavily studied under the name of "coking".

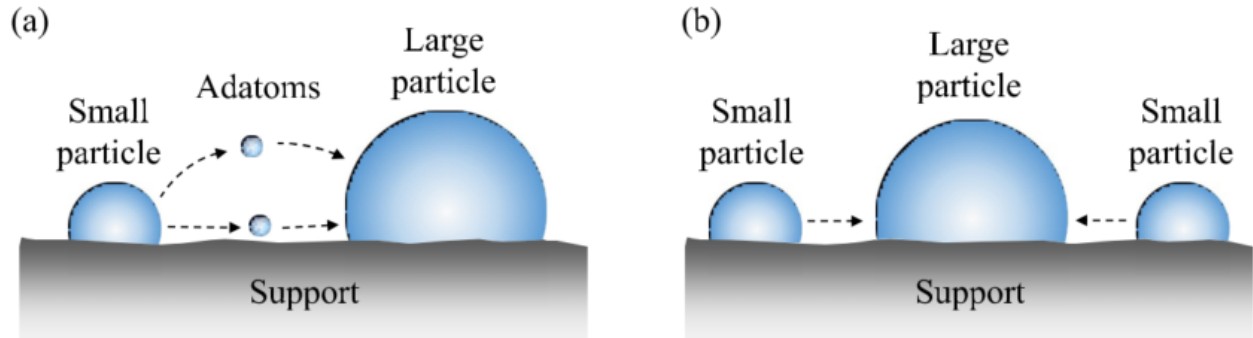

**Figure 7.** Conceptual models representing main mechanisms of metal particles sintering (via adatoms migration (**a**) or small particles agglomeration (**b**)) due to thermal degradation. Ochoa et al. [17].

### 2.3.2. Catalyst Deactivation by Coking

According to numerous works that studied coke formation and nature with several types of catalysts, coke is defined as a solid carbonaceous compound with no heteroatoms [14–16,75]. Coke nature can therefore vary from alkanes or alkenes to cyclic and aromatic molecules depending on coke formation advancement [75,76]. An average coke molecule consists in polycyclic aromatic hydrocarbons interconnected by aliphatic bridges. Its composition may be represented as $C_nH_m$, with $m/n$, or H/C ratio, usually being between 0.2 and 1.5. Coke is usually described with this H/C ratio, defining the nature of its average component. High H/C ratio is mainly aliphatic hydrocarbons, such as co-oligomers and polymers, while more condensed molecules such as polyaromatics have lower H/C value. These two "classes" of coke are respectively designated as "light" or "soft" coke and "heavy" or "hard" coke. This notion has been introduced to describe the different behaviors of coke according to its nature. The chemical structure of coke formed in catalytic processes varies with the type of reaction, catalyst, and reaction conditions. This type of deactivation is the most difficult to investigate because coke matter is constituted of multiple products from secondary reactions that contain variable amounts of carbon and hydrogen. The deactivation effect of coke is not the same according to its content, its location on the catalyst surface, its morphology and its chemical nature, which are the four main features used in literature to describe coke properties and its effect on catalytic activity. It is consequently important to understand the composition of the carbonaceous compounds in order to deduce the implied deactivation effects of the various coke molecules, but also to study the involved formation rates, mechanisms and affecting parameters. Different studies investigated coke formation in catalytic pyrolysis of plastics determining the effect of operating parameters [71,77–79].

Deactivation via coke formation is a complex phenomenon combining successive physical and chemical interactions. In catalytic processes involving hydrocarbon feedstock, coke formation usually starts with strong chemisorption as a monolayer of coke precursor or physisorption in multilayers over active sites. As illustrated in Figure 8, this results in a partial hindering of active sites (i). As carbonaceous compounds accumulate, other deactivating phenomena can occur: total encapsulation of the active site, making it inaccessible to the reactants (ii), and plugging of pores of the catalyst, blocking the access to free active sites in inner pores (iii). In advanced coke growth stages, the apparition of filamentous coke can cause changes or even disintegration of catalyst structure (iv). In addition to these chemical steps, formation of coke molecules also requires retention within the pores or on the outer surface of the catalyst. This retention may be due to trapping (steric blockage), strong chemisorption on active sites and confinement in the pores but also low volatility or solubility. The following sections provide more details about molecular mechanisms and kinetics involved in coke formation.

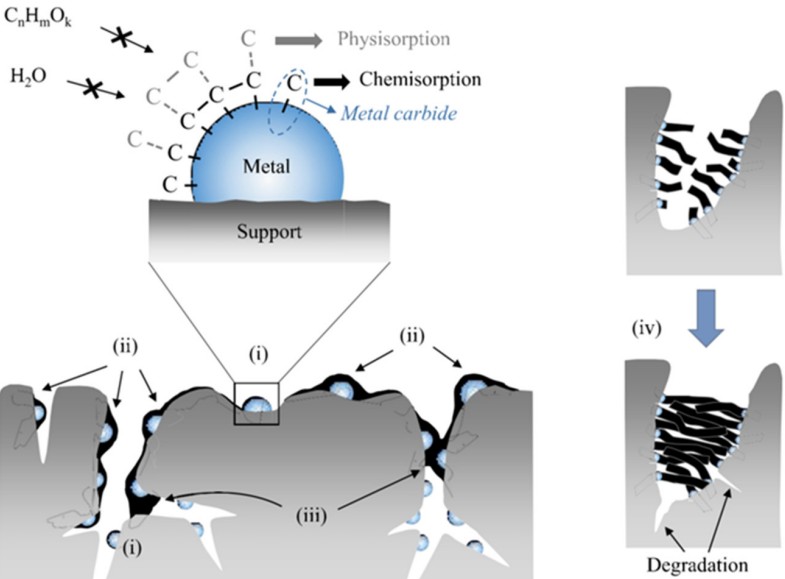

**Figure 8.** Deactivation pathways by coke formation. Ochoa et al. [17].

- **Coke formation mechanisms**

Over the years, intensive researches have been carried out to understand the mechanisms leading to the formation of complex coke molecules [75,76]. In reactions involving hydrocarbons, coke may be formed on both active sites and noncatalytic supports. Due to its important stability, coke formation is catalyzed by acid sites. As mentioned in the previous section, coke formation starts with the adsorption of coke precursors, typically olefins or light aromatics depending on the nature of the reactants. As reaction proceeds, these precursors will further react with other molecules. Coke formation involves many steps with intermolecular and intramolecular reactions. Distinction is made between low and high temperature coke. At low reaction temperatures (under 200 °C), mostly condensation and rearrangement reactions occur with the oligomerization of coke precursors. The resulting coke is mainly co-oligomers and polymers with high H/C ratios. This "light" coke formation is often reversible under specific operating conditions, making the concentration of condensation products limited by thermodynamic equilibrium. Therefore, in absence of reactant mixture, light coke molecules are retransformed to their initial compounds, referred to as "reversible coke". At high temperatures (over 350 °C), carbonaceous molecules undergo additional reactions such as hydrogen transfer and dehydrogenation, leading to the formation of polyaromatic components. This "heavy" coke is much more difficult to remove because of its high stability and its imposing size, causing steric blockage. At intermediate temperatures, a mix of these different mechanisms is observed, as presented

in Figure 9 [75]. Indeed, as coking proceeds, primarily formed light coke can undergo intramolecular condensation reactions. Carbocation intermediates that can be produced on catalyst acid sites can consequently form aromatics via dehydrogenation and cyclization reactions. These aromatics can then further react to polynuclear aromatics, which ultimately condense as coke molecules. The formation of polynuclear carbocations not only lead to the production of coke molecules but also are relatively stable, meaning they can sustain growth of molecules for quite long periods until a termination reaction occurs. At advanced coking stages, heavy polyaromatic structures are observed and can lead to both encapsulation and filamentous coke mechanisms as aforementioned.

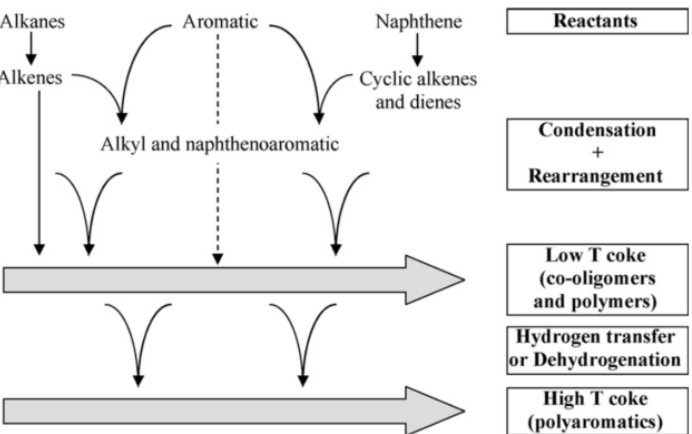

**Figure 9.** Simplified scheme of coke formation from hydrocarbons and molecular coke over acid zeolite catalysts. Guisnet et al. [75].

The main conditions and parameters for coke formation are summarized in Figure 10. The features of the reaction, the operating conditions and the studied catalyst are the main parameters determining the composition, location and rate of coke formation and therefore the involved mechanisms.

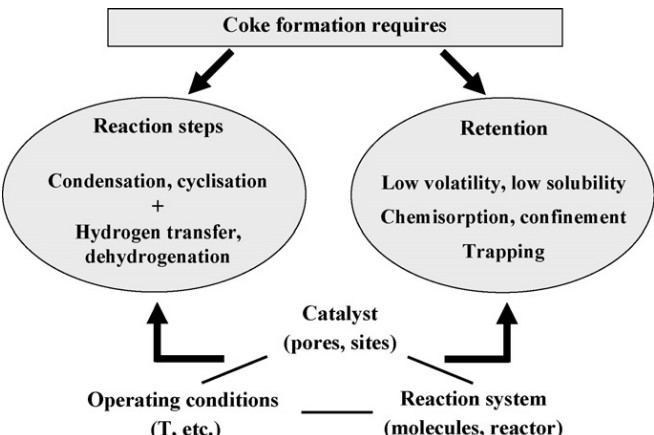

**Figure 10.** Necessary conditions for coke formation over catalysts and major influencing parameters. Guisnet et al. [75].

- **Kinetics of coking and deactivation**

Kinetic studies of coke formation are carried out using thermogravimetric analysis which provides mass evolution monitoring of the catalyst sample during the reaction. The increase of coking rate with coke content during early stages of the reaction suggests that coke formation is an autocatalytic reaction, which is coherent with the described mechanisms consisting in initial adsorption of coke precursors, leading to further reactions

to form coke molecules via rapid steps [80]. This type of analysis leads to the determination of coking rates thanks to the relation between coke content in catalyst and process time. The typical profile for the coke formation reaction rate is presented in Figure 11: the high initial coke formation rate is due to the occurrence of both catalytic and radical mechanisms, then when all active sites are blocked, the coking rate becomes approximately constant as only radical coke is formed [81]. Coke content and catalytic activity are generally compared to better understand the effect of coke over catalyst [82]. It is important to notice that the kinetics of deactivation are not necessarily proportional to coke content on the catalyst. Indeed, this depends on the selectivity of the coke formation and on the deactivation pathway. This observation is due to the variability of coke toxicity ($T_{ox}$), which could be defined as the number of active sites being inactive due to the action of one coke molecule. When a coke molecule is chemisorbed and blocks a single active site, the value of $T_{ox} = 1$. Deactivation can be limited if reactant interaction with an active site leads to coke molecule desorption ($T_{ox} < 1$) or more important if the molecule is big enough to interact with several active sites ($T_{ox} > 1$). In this case, deactivation is similar to a poisoning mechanism. However, coke can also lead to fouling or blockage, in which case a single coke molecule can block access to all active sites present in a pore or in a channel ($T_{ox} \ggg 1$). Therefore, residual activity during coke deactivation cannot be directly determined from coke content. Coke formation is highly dependent of the reactants, used catalyst and operating parameters as developed in the following section. Many studies aimed to develop kinetic models of coke formation for particular applications. The usual approach consists in splitting global coke formation phenomenon into successive elementary steps based on different hypothesized mechanisms dependent of the studied reaction [83–85]. The difficulty of developing such a model is the very large number of possible reactions having their own kinetic constant. A reaction scheme can be used to represent schematically all the possible reactions leading to the formation of coke. The reaction scheme developed by Moustafa et al. to represent coke formation during catalytic cracking of Vacuum Gas Oil is presented in Figure 12 [84]. In order to simplify the models, the number of needed kinetic parameters is substantially reduced by predicting the most important pathways of reaction and by analyzing the favorable conditions for coke formation. For instance, the kinetic model for coke formation during ethane cracking developed by Wauters et al. uses ethane, ethyne, propene, and propyne as coke precursors, reacting with gas-phase radicals H, $CH_3$, $C_2H_5$ and $C_3H_5$ [83]. The combination of this type of coke formation model with existing models for the studied reaction provides a prediction with high precision of the reaction rate, the catalyst deactivation but also the effect of operating parameters, which are discussed below [84,85].

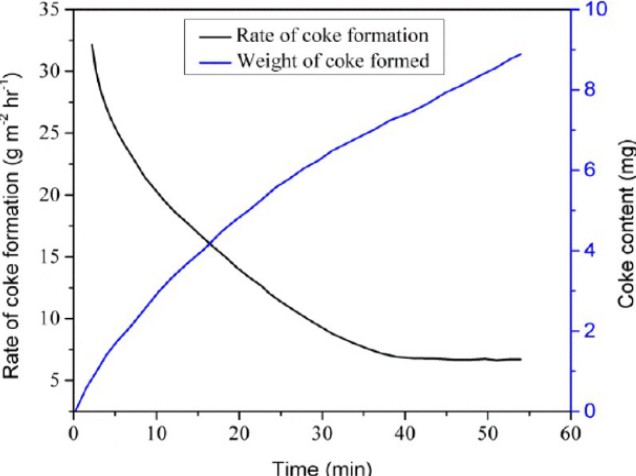

**Figure 11.** Rate of coke formation and weight content over steel cylinder during thermal cracking of ethane between 750 and 850 °C. Mahamulkar et al. [81].

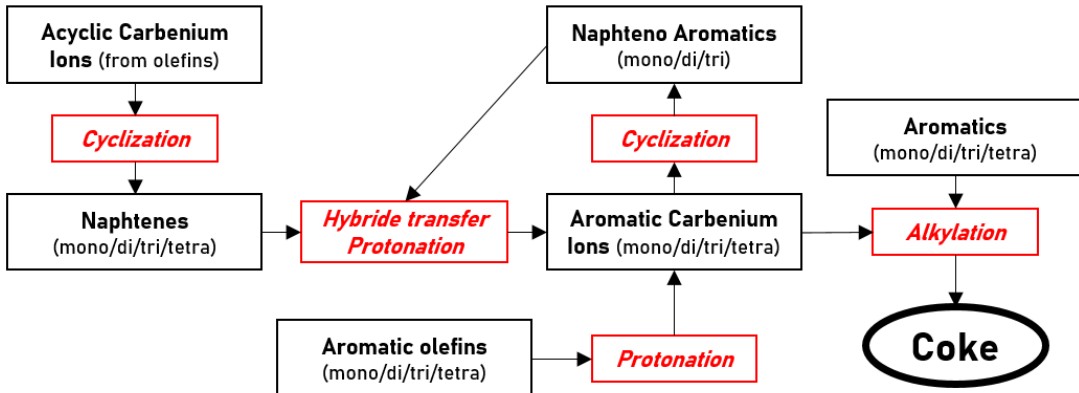

**Figure 12.** Reaction scheme for coke formation during catalytic cracking of VGO. Adapted from Moustafa et al. [84]. Black: chemical compounds. Red: reaction mechanisms.

### 2.3.3. Parameters Influencing Coke Formation

- **Reaction system**

  As detailed in the previous section, the formation of coke is the result of rearrangement and condensation reactions. Some particular molecules commonly referred to as coke precursors initiate coke formation by undergoing further reactions [86–88]. The nature of coke precursors differs according to the studied reaction and is dependent of reacting phase. Coke maker molecules can be the reactant itself, intermediates or desired products [88]. Coke precursors can be formed from light unsaturated species, such as alkenes, but also from heavier compounds such as olefins, benzene, and benzene derivatives or even polyaromatics. The formation of coke makers from these molecules is very slow due to their low reactivity and is therefore the rate-determining step of coke formation. Both retention within catalyst pores and reactivity with catalytic surface have to be satisfied and sufficient to allow initiation of coke precursors [76]. As the nature of reactants and catalysts used in the process are known, the coking behavior of a reaction system is predictable [89]. Generally, precursors from short chain alkenes and dienes undergo very fast condensation reactions leading to polar products that are easily retained on the active sites of the zeolite. On the other hand, polyaromatic precursors reactivity is not very high, but these compounds are polar enough to be retained over acid zeolites. The affinity of coke precursors with catalytic surface has a great impact on the coking behavior of a particular reaction system. Investigating the deactivation of HZSM-5 zeolite during bio-oil cracking, Guo et al. observed that coke precursor nature is different between external and internal surface, suggesting that catalyst structure impacts coke formation mechanisms [87]. The influence of catalyst structure and composition is discussed in the next paragraph.

  As coke precursors are often intermediates or desired products, coking appears as an inevitable phenomenon. All the different features of the reaction system have an impact by influencing the rate of the different possible reactions that reactants, intermediates, and by-products may undergo leading to coke formation. Deactivation studies for specific processes are usually carried out to study coke formation effects over catalytic activity loss, involved mechanisms and the influence of operating conditions in order to provide solutions to limit catalyst deactivation by coke deposition [75].

- **Operating conditions**

  As for every process involving chemical reactions, temperature influences the equilibrium and kinetics of the reaction rates but also the thermodynamic and diffusion phenomena. Temperature therefore affects both the reactivity and retention necessary for coke formation. Considering a system where feedstock contains poorly reactive coke precursors, coke formation increases with temperature as it favors the formation of intermediate coke makers. It is commonly accepted that higher temperatures enhance coke formation [90]. Analyzing coke characteristics leads to determining whether coke retention is due to low

volatility, comparing the normal boiling point with the reaction temperature, or to steric blockage, comparing molecular coke size with pore aperture. It has been proved that, at low temperatures, retention is due to low volatility of coke compounds, whereas at high temperatures, its results from their trapping within micropores [91]. As temperature has a significant effect on coke nature, it is usual to classify coke into low temperature and high temperature cokes: whatever the nature of reactants, polyaromatic coke cannot be formed at low temperatures as only condensation and rearrangement reactions are possible. On the other hand, at high temperatures, the occurrence of hydrogen transfer and dehydrogenation reactions leads to the formation of large amount of polyaromatic molecules (low H/C ratio), which composition is practically independent of the reactant and is mainly determined by the catalyst structure [76].

Reaction time is also influencing coke formation since long contact time with deactivating species will ensure extensive growth of coke structures, forming long olefin chains or increasing aromatic core number in polyaromatics. At intermediate temperatures, sufficient time of exposure leads to cyclization reactions and initiation of polyaromatic coke formation normally occurring at high temperatures. As coking proceeds, there is an accumulation of carbonaceous content over the catalyst surface, being measured by its coke content (expressed in %Carbon). Average coke nature gradually shifts from light to heavy compounds [92].

- **Catalyst structure and surface**

In terms of coke formation on zeolite catalysts, there has been extensive research carried out on the topic, including some significant findings [93–95]. The process of zeolite deactivation via coking is predominantly influenced by the pore structure of the catalyst responsible of heavy aromatic clusters. Indeed, catalyst geometry affects coke location and size since the aperture of the pores and width of channels will affect the diffusion and accessibility of coke precursors within the catalyst framework [90,96,97]. In most cases, initial molecules are relatively small and can diffuse within the catalyst where they are chemically retained and react to form coke molecules. The growth of coke molecules is limited by pore dimension, varying according to the type of catalyst (micro and macropores). The formed heavy polyaromatic coke is too imposing and is retained in the pore due to steric blockage, causing pore plugging. In industrial processes, pore size and structure have been determined to be more influential than the acidic properties of the catalysts [16]. Hence, coke formation is qualified as a shape-selective process. Coking is controlled by diffusion limitations depending on how film mass transfer and pore diffusional resistance affect the reaction of interest, but also the secondary deactivating reactions [93]. Consequently, coke yield varies within the catalyst pores and along the catalyst bed depending on the reactor dimensions and shape. However, nature and reactivity of catalyst surface also affect the coking rate. Catalyst acidity, indicated by Si/Al ratio, influences the various successive chemical steps implied during coke formation [95]. The concentration, strength and proximity of the acid sites impact coke precursor reactivity, mostly during early stages of coke formation. The quantification of the influence of these parameters is a challenge because of the difficulty to obtain zeolites with acidities and pore structures varying separately. Nevertheless, it is expected that a stronger acidic state implies faster chemical steps and stronger retention of coke molecules and precursors. Besides, higher density of acid sites leads reactant molecules to undergo more successive chemical steps along the diffusion path within zeolite crystallites, promoting condensation reactions. Strength, density, and the number of acid sites consequently enhance coking rate [75]. Catalyst deactivation by coke is in most cases a reversible phenomenon and deposited carbonaceous compounds can be removed. Regenerating processes have been investigated over the year and the different existing methods are presented in the following section.

## 3. Regeneration of Zeolite Catalysts Deactivated by Coke

For combined economic and environmental concerns, processes for catalyst regeneration have been investigated over the years. Coke removal is achieved via three methods:

oxidation, gasification, or hydrogenation. Each process has its advantages and drawbacks according to the catalyst type to be regenerated and to the nature and structure of coke. Particular attention is given to the regeneration of zeolite materials commonly used during catalytic pyrolysis of plastic wastes.

### 3.1. Coke Combustion with Air/Oxygen

The most frequently used method to regenerate coked catalysts in the industry is coke combustion using air or oxygen. Spent catalysts are usually placed in a fixed-bed reactor and are treated with oxygen-containing gas at a high temperature. While nitrogen is mostly used as a diluent in laboratory-scale tests, steam is used in full-scale plant operations [15]. Coke removal reaction with air or oxygen is a rapid process occurring at moderate-high temperatures (usually around 400–500 °C). Coke combustion with oxygen is used for catalyst regeneration of common industrial processes such as cracking or catalytic reforming [14–16].

#### 3.1.1. Oxidation Mechanisms

Coke oxidation with oxygen is an exothermic reaction that is usually described by the following set of equations, representing the total oxidation of solid unspecified carbonaceous compounds, where H(s) represents hydrogen atoms attached to solid coke compounds [23]:

$$2H(s) + \frac{1}{2}O_2\,(g) \rightarrow H_2O(g) \qquad\qquad -121.0 \text{ kJ/mol}, \tag{1}$$

$$C(s) + O_2\,(g) \rightarrow CO_2\,(g) \qquad\qquad -395.4 \text{ kJ/mol}, \tag{2}$$

$$C(s) + \frac{1}{2}O_2\,(g) \rightarrow CO\,(g) \qquad\qquad -110.4 \text{ kJ/mol}, \tag{3}$$

$$CO(g) + \frac{1}{2}O_2\,(g) \rightarrow CO_2\,(g) \qquad\qquad -285.0 \text{ kJ/mol}. \tag{4}$$

From the determined reaction enthalpies, it can be shown that the exothermicity of the process is mostly due to oxidation of carbon, even though energy emission of hydrogen oxidation is not negligible. From Temperature-Programed Oxidation (TPO) experiments, it has been shown that light coke is primarily oxidized, as hydrogen oxidation is easier, leading to light coke dehydrogenation forming heavier coke. High carbon content coke is burnt off afterwards at higher temperatures [98,99]. This is why water is the first molecule observed on a classical TPO spectrum, followed by CO and $CO_2$ production, which corresponds to the competing mechanisms of carbon oxidation [98,100,101]. Intrinsic mechanisms of coke have not been formally determined yet since coke nature has a great influence over the oxygenated intermediates formed during oxidation, which are considered as precursors for carbon oxide formation [102]. The following equations presented in Table 1 are commonly used to describe more precisely the steps of coke combustion, from light coke dehydrogenation to heavy coke oxidation via the formation of oxygenated intermediates [100,101].

**Table 1.** Overall mechanism for light and heavy coke oxidation with oxygen.

| | | |
|---|---|---|
| **Light coke combustion** Hydrogen oxidation | $-C_nH_{2n} + \frac{n}{2}O_2 \rightarrow -C_n + nH_2O$ | (5) |
| **Heavy coke combustion** Carbon oxidation | $-C_f + O_2 \rightarrow -C(O_2)$ | (6) |
| | $-C(O_2) + -C_f \rightarrow -C(O) + CO$ | (7) |
| | $-C(O) \rightarrow CO$ | (8) |
| | $-C_f + O_2 \xrightarrow{-C(O)} CO_2$ | (9) |
| | $-C(O_2) \rightarrow CO_2$ | (10) |

In these equations, $-C_nH_{2n}$ and $C_n$ represent light coke and hard coke, while $-C_f$ is used to describe a free carbon site available for dioxygen chemisorption. The latter ensures the formation of dissociated $(-C(O))$ and undissociated $(-C(O_2))$ surface oxides turning into the corresponding molecular gases once atom binding is completed. The competition between rearrangement (8) and desorption (10) of the formed complexes is responsible for the complexity of the coke oxidation process. The latter is usually qualified as completely achieved when $CO_2$ is formed while partial oxidation leads to CO formation. As for every reaction, coke oxidation is influenced by different parameters (as developed in Section 3.1.3. [101,103].

The evolution of reactivity is studied following the composition of oxidation products, especially the CO/CO2 ratio, to monitor which reaction is favored. Overall, coke oxidation mechanisms are governed by combustion temperature and coke nature: the more the coke is condensed and "heavy", the more the temperature needs to be high for the oxygen to oxidize solid carbon compounds [103]. Coke location and therefore catalyst geometry are also determining parameters as they govern the diffusion of oxygen within catalyst pores and the accessibility of coke. The process of coke oxidation is consequently often qualified as a shape-selective process [93]. However, this problem is more related to mass transfer and diffusion limits than to reaction mechanisms and will be further developed in a following section (Section 3.1.2).

### 3.1.2. Reaction Model: Kinetics and Mass Transfer

- **Kinetics**

Determining reaction rates is important for process design and optimization to avoid the apparition of hot spots leading to catalyst attrition. Consequently, over the last decades, many studies worked on the determination of kinetic parameters for coke combustion applied to various catalysts and deactivating reactions. Two approaches are possible for kinetic study completion: coke oxidation can be considered in its overall form, thus considering a global reaction rate for the direct combustion of coke to carbon oxides. Otherwise, intrinsic coke burning reactions may also be considered to approach hypothesized intermediate mechanisms occurring at the reaction site within the catalyst [104]. As the results from both approaches are similar, the determination of a global reaction rate for the overall coke combustion is usually used for process design needs as it provides a sufficient approximation to study oxidation kinetics. Models considering intrinsic reactions are based on various assumptions aiming to describe reality as close as possible. Complexity of coke compounds leads to a limitation for coke oxidation modelling as the reaction rate is correlated with numerous varying parameters such as coke and catalyst nature and structure, among others [93,103,105,106]. While kinetic studies only focused on the carbon oxidation rate, some works noticed that hydrogen oxidation effect in oxidation exothermicity was not negligible in the early stages of combustion and pointed out the importance of integrating the hydrogen reaction rate in kinetic studies, especially to evaluate the risk of sintering related to the apparition of hot spots [98]. The multiple-reaction model detailed in Table 2 is based on the overall oxidation reaction and offers a relatively simple description of the coke burning process [107,108].

**Table 2.** Stoichiometry and overall reaction rates for solid carbon and hydrogen oxidation.

| Stoichiometry | Kinetics | | Global Reaction Rates |
|---|---|---|---|
| | | | *[Hydrogen]* |
| $2H(s) + O_2(g) \rightarrow H_2O(g)$ | $r_1 = k_{Hs}C'_{H0}P_A$ | (11) | $r_{H_c} = k_{Hs}C'_{H0}P_A$ |
| $C(s) + O_2(g) \rightarrow CO_2(g)$ | $r_2 = k_2C'_CP_A$ | (12) | *[Carbon]* |
| $C(s) + O_2(g) \rightarrow CO(g)$ | $r_3 = k_3C'_CP_A$ | (13) | $r_{C_c} = (k_2 + k_3)C'_CP_A = k_CC'_CP_A$ |

$k_{Hs}$ and $k_C$ are the rate constants for hydrogen and global carbon oxidation, while $k_2$ and $k_3$ are the separate rate constants for carbon oxidation to $CO_2$ and CO. $C_C'$ and

$C_{H0}'$ represent the carbon content and the initial hydrogen content in coke, while $p_A$ is the oxygen partial pressure. This simplified model gives a reliable representation of the coke burning process, where the equations are expressed as first order with respect to the oxygen partial pressure and reactants concentration. Determination of kinetic parameters confirmed that hydrogen is oxidized faster than carbon, thus "soft" coke is firstly oxidized and turned into "hard" coke, which is more difficult to oxidize. Both hydrogen and carbon reaction rates were correlated with the Arrhenius equation determining the temperature effect over kinetic constants and activation energies for specific applications. Indeed, as coke nature and catalyst structure vary from a process to another, it has been proved that mass transfer within the catalyst mostly limits coke oxidation. Therefore, kinetic parameters vary for each application, requiring a different study for each process. The importance of diffusion and mass transfer for coke oxidation will be discussed in the next section.

- **Diffusion and mass transfer**

Being a fast equilibrium reaction, coke oxidation of both carbon and hydrogen in coke is mainly controlled by oxygen diffusion within the pores of the inert catalyst pellet towards coke reactive surface. Consequently, coke is firstly removed from the edge of the pellet and oxidation progresses to the center core of the pellet as oxygen diffuses farther into the solid catalyst matrix. This model of diffusion, known as shrinking-core model (SCM), is used to describe heterogeneous reactions where a gas-phase reactant reacts with species contained in a porous solid material. SCM provides a mathematical representation of gas-phase diffusion, here oxygen, throughout the catalyst and its reactivity with coke [109]. Using diffusivity and molar balance equations finally offers an ideal prediction of oxygen concentrations across the catalyst pellet at various times. This model implies catalyst particle to be divided into two distinct regions, a non-reacted carbon-rich core at the center of the pellet surrounded by a carbon-free outer shell where coke removal is achieved. The delimitation of these zones is the reaction interface where oxidation exclusively occurs, which is not representative of the real phenomenon. As coke oxidation advances, this boundary moves towards the center of the grain as illustrated in Figure 13a.

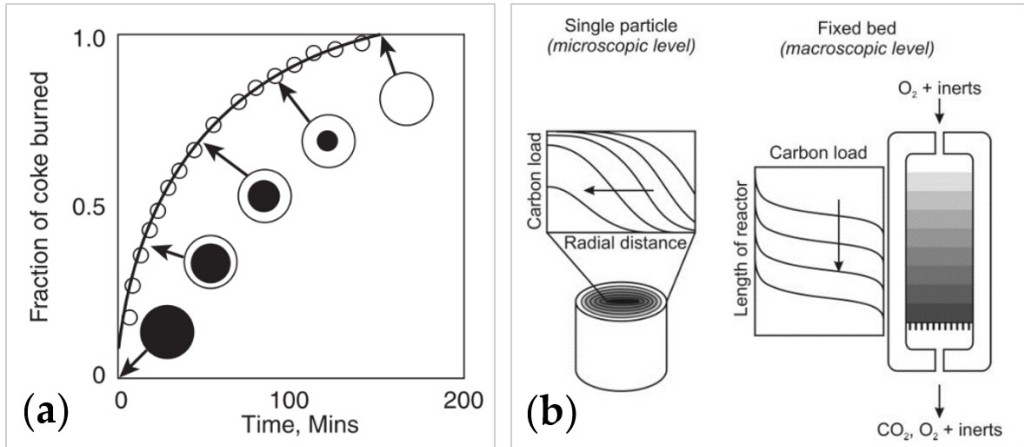

**Figure 13.** (**a**) Shell progressive regeneration of fouled pellet following shrinking-core model (SCM). Richardson et al. [109]; (**b**) Coke burn-off during the regeneration of coked catalyst pellets in a fixed bed reactor. Müller et al. [110].

Even though SCM provides a good estimation of coke oxidation progression within a single catalyst grain, some assumptions lead to a lack of accuracy representing reality. For instance, in this model, reaction rates are ignored and coke oxidation is considered immediate at the unreacted coke external surface. During the real oxidation process, all oxygen does not stop instantaneously to react at the unreacted core surface and pursue its way within the coked pores, generating the apparition of a "partially" oxidized zone where

carbon content decreases from $C_0$ (unreacted core carbon content) down to a carbon-free region, as presented in Figure 13b [110]. Kern et al. completed the existing shrinking-core model (SCM) including the influence of pore diffusion as well as the intrinsic kinetics to model more precisely the effective rate of coke oxidation [111]. Some studies adapted this work on a single catalyst grain to a fixed-bed regenerating reactor, modelling carbon concentration profiles alongside the deactivated bed. As expected, by transposition from the grain scale to the bed scale, when oxygen is fed by the top of the reactor, it was proved that a greater coke removal is achieved in the upper fraction of the fixed bed as it is more exposed to oxygen flow, while on the other hand, carbon-rich coke remained in the bottom fraction. Therefore, coke oxidation is controlled by oxygen diffusion within both a single catalyst particle (microscopic) and a fixed-bed reactor (macroscopic) [108,110–112].

### 3.1.3. Parameters Influencing Coke Formation

- **Catalyst structure and composition**

Coke oxidation, being a shape-selective process where the reaction rate is limited by oxidant diffusion, the catalyst structure appears as an important parameter since the geometry of the internal pores will influence the mass transfer within the catalyst. Several works demonstrated the correlation between catalyst structure and coke removal rate, mainly related to the variable coke accessibility to oxygen due to the difference of pore structures according to the catalyst geometry [78,91,93,113]. Indeed, following the size and aperture of pores, coke location will change and will deposit preferentially on the outer or internal surface, being more or less accessible to oxygen. Therefore, while coke on the outer surface will be easily removed, oxidation of inner compounds will be more difficult and dependent of oxygen diffusion within the pores. As an illustration, easier coke removal is observed on a HFAU-type catalyst at 550 °C while a temperature of 600 °C is necessary for coke removal from HEMT zeolite, regardless of framework composition and coke content. [91]. This has been attributed to the preferential deposition of coke on the outer surface because of the small pore apertures of HFAU structure, making coke more accessible for oxidation. Similar results were observed by Magnoux et al. comparing oxidation of coke formed on HY, H-mordenite and HZSM-5 zeolites during n-heptane cracking. In addition to the faster oxidation of coke deposited on the outer surface, the influence of internal channels and pore size for oxygen diffusion has been proved: macropores, supercages, and interconnected channels offer a better circulation of oxygen, subsequently affecting the contact between oxygen and coke deposits located over the inner surface [93,113].

Zeolite framework as well as the number and strength of active sites are also implied in catalyst regeneration and coke removal efficiency. Influence of catalyst composition over coke oxidation is observed on both metal and zeolite catalysts since both the metallic and acidic acid sites participate in the coke removal process. Interactions between oxygen and the catalyst may vary according to the composition of the active sites or even its support. Moljord et al. observed that a high density of framework aluminum atoms facilitates coke oxidation over HY zeolites with Si/Al ratios from 4 to 100 [114]. Another study proved that, over a Pt-Sn/$Al_2O_3$ catalyst, oxygen is activated by platinum particles rather than alumina support, therefore coke located over metallic Pt sites is preferentially oxidized [115]. Consequently, knowledge of catalyst composition is also important to understand coke oxidation mechanisms since nature of deactivated surface has also a role in coke removal rate as it determines intrinsic mechanisms of oxidation. Even though diffusion is the limiting step of the reaction, the nature of deactivated surface has also a role in coke removal rate as it determines the intrinsic mechanisms of oxidation. Due to the variety of catalysts, different activation step mechanisms for the reaction of an aromatic core in coke are proposed. A mechanism suggested by Dong et al. suggests that the initiation step for coke oxidation involves the formation of radical carbocations from any accessible aromatic cores of coke molecules. This mechanism, presented in Figure 14, is commonly accepted as the main reaction for coke oxidation [116]. The relative acidity of the coke-carrier surface accounts for the existence of preferential coke removal sites, as the

formation of carbocation intermediates from the deposited coke are more or less favored. Therefore, oxygen reactivity is not only determined by the nature of coke itself but also by the composition of the deactivated surface.

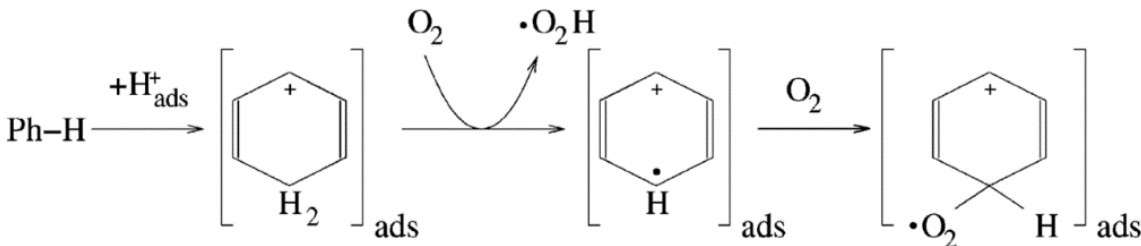

**Figure 14.** Carbocation mechanism for the reaction of oxygen with an adsorbed coke molecule (presented by Dong et al. [116] and adapted by Keskitalo et al. [102]).

- **Operating parameters**

For process optimization purposes, the influence of operating parameters on coke oxidation efficiency has been heavily studied. The main four parameters that can be easily modulated at an industrial scale are temperature, time on stream, oxygen concentration, and flowrate. While oxygen is diluted in $N_2$ in laboratory experiments, industrial processes often use a steam and air mixture. In classical conditions, coke oxidation is operated in the temperature range 400–600 °C and with the oxygen concentration in gas from 0.05 to 10 vol%, with a time of exposure varying according to the degree of coke removal to be achieved [117]. For example, during the regeneration of HZSM-5 catalyst coked during ethylbenzene conversion, Jong et al. observed that 67% of coke was removed after 0.5 h and 93% after 2 h, while 6 h are needed to achieve complete coke removal [118]. As the oxidation reaction rate is correlated with Arrhenius equation, higher temperatures could accelerate the reaction to achieve better coke removal. However, temperature is limited by the catalyst material, which could be altered in case of excessive temperature exposition. The operating temperature can therefore vary according to the deactivated catalyst and is set at the maximum value allowed by the catalyst to increase coke removal efficiency, but also to ensure that the catalyst structure is not damaged. Marcilla et al. observed a structural change in HZSM-5 zeolite after a 900 °C treatment, leading to further catalytic activity loss, while a HUSY-type zeolite exposed to the same treatment recovered all its activity without any structural alteration [78]. To avoid the apparition of hot spots and catalyst sintering, the oxygen concentration in incident gas also has to be controlled. Indeed, as coke oxidation is exothermic, a too high concentration of reactant could lead to local temperature rises, which could further deactivate the catalyst. Santamaria et al. investigated the temperature behavior within a reactor during coke oxidation with different oxygen inlet strategies, as illustrated in Figure 15 [99]. Even though this experiment shows that temperature is reduced in the reactor with lower oxygen concentrations, coke removal efficiency is also greatly impacted. Therefore, a balance has to be found between thermal risk and oxidation efficiency. As the risk is mainly present at high coke contents, the combustion process is typically controlled by initially feeding low concentrations of air before increasing the oxygen concentration gradually as the carbon content decreases [119]. Coke oxidation, being limited by mass transfer and following the shrinking-core model, coke removal increases with time and complete regeneration is theoretically achieved at infinite time. Time of exposure is a parameter determined to comply with industrial needs in terms of coke removal and catalytic activity recovery. For economic reasons, optimization studies of oxidation processes tend to maximize catalytic activity recovery, which is correlated to coke removal, and to minimize time on stream by variation of all the aforementioned parameters [120].

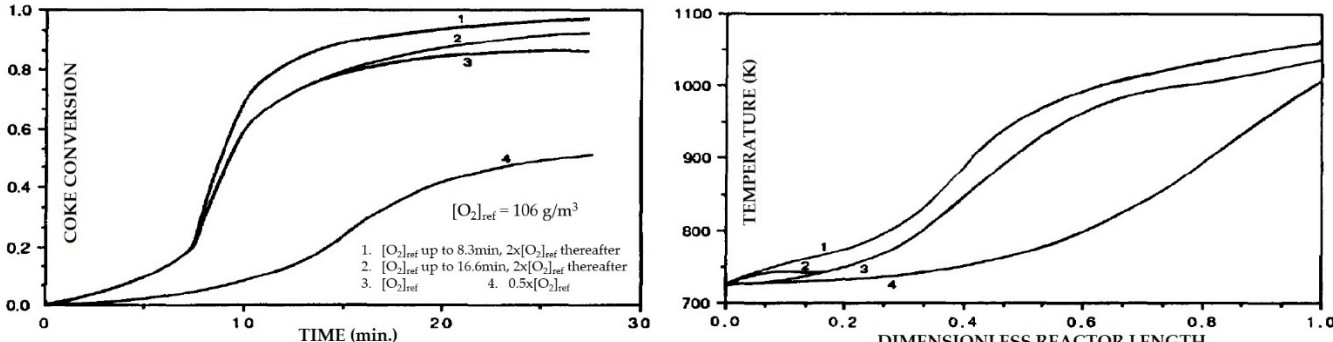

**Figure 15.** Average coke conversion and temperature profiles in the reactor for different oxygen inlet concentration strategies. Adapted from Santamaria et al. [99].

### 3.1.4. Limitations of Oxygen Oxidation

The main drawback to coke oxidation with air or oxygen is the temperature limitation used to avoid irreversible damage to the catalyst and a loss of catalytic activity after regeneration. Among the principal possible structural changes due to thermal degradation, dealumination has been studied in zeolite materials for its capacity to promote the formation of mesoporous systems, giving an interest to determine the reaction rate and influencing parameters [121,122]. In some particular cases, dealumination turns out to be a way to improve catalytic activity due to the modification of acidic properties in the structure [123]. However, structural changes during a regeneration process are not desired as it often causes a decrease of active sites by sintering or damages on the crystallinity of the catalyst framework, leading to an irreversible loss of catalytic activity [78,124].

Intensive work was carried out to achieve the coke oxidation reducing temperature of the regeneration process, therefore avoiding thermal damage. A well-known technique is catalyst improvement with metal impregnation, offering both inhibition of heavy polyaromatic coke and temperature reduction during regeneration [125]. Moreover, it was shown that coke removal close to the metal is easier, with lower temperatures and shorter exposition times, suggesting that coke oxidation is promoted by the presence of metal particles [126]. However, coke combustion also has the disadvantage to not guarantee catalyst stability after several regeneration cycles. Different studies have shown a significant catalytic activity change or loss compared to the fresh catalyst after repetition of deactivation/regeneration cycles [127]. Lu et al. observed a loss of catalytic activity during catalytic cracking of toluene over $Ni/\gamma\text{-}Al_2O_3$ catalysts after successive regeneration cycles, as illustrated in Figure 16 [128]. Therefore, metal-impregnation method may not be suitable for long-time reuse of catalysts [129]. They attributed the loss of surface area to the sintering of metal particles and to the remaining coke. In order to avoid the issues related to coke combustion, the use of an oxidant other than oxygen has also been investigated to achieve improved catalyst regeneration in milder conditions. These alternative methods for oxidative processes are discussed in the following sections.

### 3.2. Advanced Oxidation Processes (AOP)

Advanced Oxidation Processes (AOPs) can be used for the regeneration of adsorbent materials used for wastewater and gas treatments [130–134]. Their application to coke removal from heterogeneous catalysts gained recent interest in order to develop new alternative methods for coke oxidation processes. Indeed, coke combustion with air/oxygen has some limitations as it can cause further catalyst deactivation due to thermal degradation. AOPs are based on the insight formation of strong oxidizing chemical species leading to coke removal in milder conditions. The different oxidizing species are presented in Table 3 [135]. Due to its strong redox potential, the formation of OH radicals is desired among all the possible oxidants susceptible to be present. The different existing methods for advanced oxidation that have been used for coked catalyst regeneration are presented

in the following paragraph. Because of the recent interest in AOPs for coke removal over the catalysts, few studies investigated these alternative regeneration methods. According to literature, AOPs may be classified into three main groups: ozone-based processes, photocatalytic processes, and Fenton reaction-based processes [136].

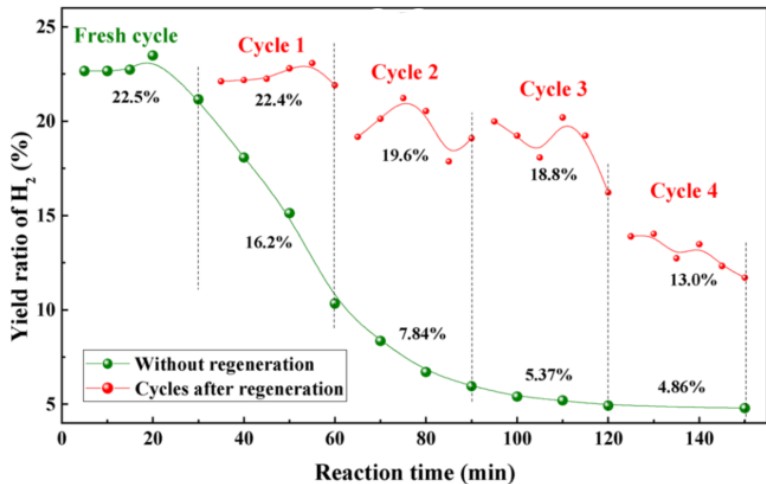

**Figure 16.** $H_2$ yield of a metal-impregnated cracking Ni/$\gamma$-Al$_2$O$_3$ catalyst after different cycles of regeneration with oxygen for 3 h. Lu et al. [128].

**Table 3.** Standard potential of oxygen, ozone, and principal oxygenated radicals at 25 °C and 1 atm.

| Oxidant | Redox Couple | Redox Potential (V) |
|---|---|---|
| Oxygen molecule | $O_2/H_2O$ | 1.23 |
| Hydrogen peroxide | $H_2O_2/H_2O$ | 1.78 |
| Ozone | $O_3/O_2$ | 2.07 |
| Atomic oxygen | $O\bullet/H_2O$ | 2.42 |
| Hydroxyl radical | $HO\bullet/H_2O$ | 2.80 |

### 3.2.1. Ozonation

Ozone has been a molecule of interest for many years due to its strong oxidizing power and is used for both industrial purposes (chemical, pharmaceutical, etc.) and environmental applications (water, soils, and air treatment). The different applications are detailed in the Rakovsky et al. review [137]. Ozone is a very unstable molecule, whose reactivity is given by the different properties of its resonance structures, providing electrophilic characteristics and oxidizing power [138]. This molecule does not exist naturally at atmospheric pressure and is therefore created artificially with different methods such as electrical discharge (ED), electrolysis, or irradiation. The first-cited technique is the most commonly used, consisting in treating air or pure oxygen with extremely high voltages (20,000 V) to form ozone. Due to its instability, ozone has a short lifetime of tens of minutes at ambient temperature and significantly lower—several seconds—at higher temperatures before undergoing thermal degradation and naturally decomposing to oxygen [139]. The thermal gas-phase decomposition of ozone leads to the formation of oxygen in the presence of a third molecule being $O_2$, $O_3$, $CO_2$, $N_2$ or other gaseous body, impacting the kinetic constant of reaction [140].

"Catalytic" decomposition can also occur when ozone is exposed to active materials such as metal oxides or zeolites. This type of degradation leads to the formation of adsorbed oxygenated species such as atomic oxygen, peroxide ions and hydroxyl radicals as described in a mechanism proposed by Li et al. [141]. The use of ozone over a catalytic surface therefore produces a great variety of species from molecular gases to short lifetime radicals, which are available for coke oxidation [142,143]. The following set of equations

describes ozone decomposition over metal oxides and zeolites. In this mechanism, which is adopted in many studies, * represents surface active sites.

$$O_3 + * \rightarrow *O_3, \tag{14}$$

$$*O_3 \rightarrow *O + O_2, \tag{15}$$

$$*O + O_3 \rightarrow *O_2 + O_2, \tag{16}$$

$$*O_2 \rightarrow * + O_2, \tag{17}$$

$$2 *O \rightarrow 2 * + O_2, \tag{18}$$

$$*O + H_2O \rightarrow 2 HO\bullet. \tag{19}$$

This mechanism illustrates the decomposition of ozone to oxygen but also emphasizes the formation of adsorbed oxygenated species such as atomic oxygen (Equation (15)). Hydroxyl radicals are also susceptible to form when close to hydrogen, due to humidity or to the presence of other species (Equation (19)) [144]. Due to the formation of these intermediate oxygenated species, ozonation can act following two different pathways: direct and indirect. In the direct mechanism, ozone is the dominant oxidizing agent via a direct electrophilic attack of the molecular ozone on organic compounds, whereas an indirect mechanism relies on the participation of the aforementioned oxygenated intermediates formed from catalytic ozone decomposition [136]. These highly reactive species can attack organic coke aggressively by either extracting hydrogen to form water, as it is the case with alkanes and alcohols, or by attaching itself to the molecule breaking double bindings, as observed for unsaturated species such as alkenes and aromatics.

Due to the recent renewed interest of ozone use for oxidation processes in gas-phase, very few studies focused on the determination of involved mechanisms and kinetics of ozonation. Their scope is mostly focused on the influence of the operating parameters over regeneration capacity. The main advantage of ozonation compared to coke combustion with oxygen is the lower range of temperature needed to achieve oxidation, lowering the risk of catalyst thermal degradation. Mariey et al. achieved coke removal from HY zeolites deactivated by cyclohexene cracking with ozone at 180 °C while 500 °C was needed for oxygen removal [145]. Therefore, coke removal over zeolite catalysts is successfully achieved at temperatures between 100 °C and 200 °C with varying parameters. The investigated ozonation conditions found in literature are summarized in Table 4. The main impacting parameters are temperature, ozone concentration, gas flow-rate, and time on stream, as well as structure and acidity of deactivated catalyst. The inlet ozone quantity is determined by two factors: $O_3$ concentration, generally a few ppm, and gas flowrate. These two parameters are correlated by the capacity of the ozone generator placed ahead of the pilot: the higher the flowrate, the lower the concentration. Some comparative studies demonstrated that a similar behavior is observed for oxygen and ozone oxidation processes [116,145–147]. Similar to coke combustion, ozonation is a shape-selective process controlled by the diffusion rate of oxidizing species within the catalyst pores. Microscopic and macroscopic coke profiles within the catalyst pellet and alongside the catalyst bed were found to be similar to oxygen-regenerated samples. Hence, ozone-based oxidation firstly reacts with external coke and progresses towards the center of the pellet and coke is firstly removed at the upper part of the catalyst bed when ozone-containing gas is fed by the top of the reactor [146,148]. Consequently, the expected effect for time is a higher coke removal with time-on-stream (tos) increase. Khangkham et al. reported the apparition of a plateau after 2 h, suggesting that coke removal is not complete [148]. Richard et al. also found that a maximum of 74.3% of coke could be removed from deactivated catalyst after 6.5 h [149]. This observation is explained by the diffusion limitations due to the instability of the created oxidizing species, causing their rapid degradation before they reach the unreacted coke surface. Despite the presence of remaining coke, some studies reported that catalytic activity may be totally recovered, suggesting that partial coking is

tolerated before deactivation [146]. Temperature plays an important role during ozonation as it is involved in the kinetics of both coke oxidation and oxidizing species degradation. The competition between these two phenomena accounts for the apparition of an optimal regeneration temperature as observed by Querini et al. during the regeneration of Y-zeolites coked during isobutane alkylation [150]. The use of ozone for coked catalysts regeneration remains relatively new and is gaining in interest. The influence of the operating factors still needs to be studied in detail. The following table summarizes the operating conditions that have been investigated for the regeneration of zeolite catalysts via ozonation.

**Table 4.** Operating conditions for coke removal from coked catalysts using ozone found in literature.

| Ozonation Conditions | Deactivated Catalyst | Deactivating Reaction | Ref. |
|---|---|---|---|
| $O_2 + O_3$ flow 120 $cm^3.min^{-1}$ (ratio not given), 137 °C. | HY and HYS zeolite | Exposition to cyclohexene at 347 °C. | [145] |
| $O_3/O_2$ mole ratio 0.05–0.06, 150 °C, 90 min. | HZSM-5 zeolites (Si/Al ratio 35 and 70) | Methanol conversion to hydrocarbons and o-xylene isomerization. | [146] |
| $O_3/O_2$ mole ratio 0.05, 200 °C, 4h (bed agitation). | HY zeolite | Exposition to various alkanes and alkenes at 500 °C. | [147] |
| $O_3$ conc. from 16 to 50 $g/m^3$, 20–150 °C, 0.5–4 h. | ZSM-5 zeolite | PMMA cracking process at 250–300 °C. | [148] |
| $O_3$ conc. from 4 to 25 $g/m^3$, 50–200 °C, 2–8.5 h. | Undefined zeolite | Not given. | [149] |
| $O_3/O_2$ mole ratio 0.01, 125 °C, 4 h + $H_2$ regeneration | Y-zeolite (UOP, Y-54) | Isobutane alkylation in liquid phase (25–80 °C). | [150] |

3.2.2. Other Methods for Advanced Oxidation

- **Non-Thermal Plasma (NTP)**

Among the alternative techniques to regenerate coked catalysts, the use of non-thermal plasma (NTP) has been proved to be an efficient technique for coke removal from zeolite materials at ambient temperature [151]. This method is based on the formation of reactive oxygenated species (ROS), such as radicals, excited atoms, ions, and molecules, thanks to the generation of highly energetic electrons in plasma discharge. Generated species from dioxygen exposition to plasma are mostly positive and negative ions ($O^{2+}$, $O^-$, $O_2^-$) but also atoms and molecules such as O-atoms or ozone, which are able to form radicals or other reactive species. This type of regeneration is carried out in a dielectric barrier discharge (DBD) reactor where spent catalysts are placed between two electrodes. The oxidative species are generated thanks to the plasma discharge from oxygen contained in the gas passing through the reactor. This type of reactor, which generates plasma at atmospheric pressure, is experimented by Hafezkhiabani et al. to perform Pt-Sn/Al$_2$O$_3$ decoking [152]. Two different geometries of DBD reactors are found in literature: point to plate (Figure 17a) and fixed-bed reactor (Figure 17b). The latter was used by Astafan et al. who studied the NTP regeneration of faujasite zeolite coked from propene transformation at 623 K and achieved complete regeneration of the catalyst at ambient temperature with a deposited power of 12 W [153]. Consequently, coke removal can be achieved thanks to the generation of these oxidizing species that are diffusing within catalyst structure and oxidize coke organic compounds. Similarly to coke combustion and ozonation, this process is highly dependent of the catalyst structure due to the diffusion limitations, as demonstrated by Pinard et al. who studied NTP catalyst regeneration over MFI, MOR, and FAU zeolites [113]. Different parameters, such as input power, gap between electrodes, gas flowrate and nature of diluent (N$_2$, He, Ar), as well as catalyst mass and compactness, have been investigated to provide deeper understanding of this process, which remains a relatively new and unknown method despite its promising aspects [154].

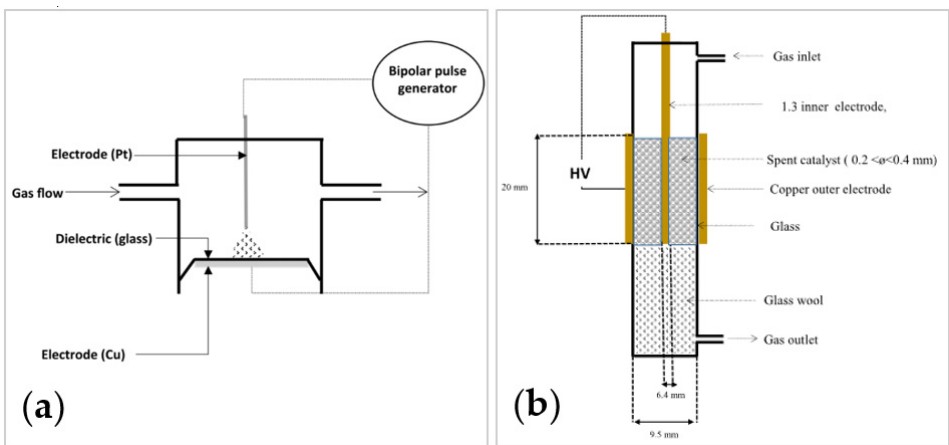

**Figure 17.** Schematic representation of (**a**) point-to-plate reactor [113] and (**b**) fixed-bed dielectric barrier discharge (DBD) reactor [153] for coked catalyst regeneration via NTP.

- **Hydrogen peroxide and OH-derived Fenton radicals**

Oxidation processes based of Fenton reaction have applications in various domains such as soil or wastewater treatments. However, its application for deactivated catalytic materials is relatively recent and, consequently, there is currently no studies mentioning application of Fenton chemistry over coked catalysts and only very few papers dealing with fouled zeolite [155–157]. These papers are not properly dealing with coke removal as defined earlier but with deactivating species referred as humins, having similar behavior with coke (adsorption, steric blockage). Application of Fenton reaction could therefore be relevant for coke removal. These processes, occurring in liquid acidic medium, are based on the generation of OH radicals from hydrogen peroxide in the presence of Fe salt, which acts as a catalyst taking part in $Fe^{3+}/Fe^{2+}$ redox cycle. The so-called Fenton reaction is described by the following mechanism:

$$H_2O_2 + Fe^{3+} \rightarrow HO_2\bullet + H^+ + Fe^{2+}, \tag{20}$$

$$H_2O_2 + Fe^{2+} \rightarrow HO\bullet + OH^- + Fe^{3+}, \tag{21}$$

$$HO\bullet + Fe^{2+} \rightarrow OH^- + Fe^{3+}, \tag{22}$$

$$HO\bullet + H_2O_2 \rightarrow HO_2\bullet + H_2O, \tag{23}$$

where Reactions (20) and (21) are desired because they lead to the formation of radicals OH, while Reactions (22) and (23) are undesired pathways. Highly reactive hydroxyl radicals are consequently produced at low temperature. Using this method, Morales et al. reported total coke removal and catalytic activity recovery below 100 °C for ZSM-5 zeolite fouled during glucose dehydration [157]. According to them, similar diffusion limitation issues are observed due to the rapid recombination of highly oxidative radicals and the selection of appropriate reaction conditions is a key factor to achieve coke removal. Oxidation mechanism can be, similarly to ozonation, direct from hydrogen peroxide or indirect with formed OH radicals. However, further research is needed to verify the efficiency of this method over "real" coke, and especially the potential Fe impurities generated by this method. Fenton-reaction-based processes may be a new alternative method for catalysts regeneration via oxidation under mild conditions.

### 3.3. Other Regenerating Reactions
#### 3.3.1. Gasification

Although oxidative treatments are very commonly used for coked catalysts regeneration, the main drawback of these techniques is the formation and emission of carbon dioxide. While the reduction of greenhouse gases emission is an actual industrial challenge,

coke combustion for spent catalysts regeneration produces almost half of the emitted $CO_2$ in an FCC unit [158]. Gasification provides an alternative method to mitigate $CO_2$ emission during catalyst regeneration, using $H_2O$ or even $CO_2$ as feedstock to remove coke at around 700–900 °C (see corresponding reactions Table 5). The latter uses $CO_2$ as an oxidizing agent (redox potential 1.33 V) reacting with coke to form carbon monoxide (Equation (20)). The equilibrium of this highly endothermic reaction favors CO production at temperatures above 700 °C [159]. Due to low $CO_2$ reactivity and the high temperatures needed, the scope of application is limited to catalysts with high resistance to heat. Otherwise, in such conditions, catalysts may suffer structural damages or sintering. Use of $H_2O$ as the reactant for gasification ensures the direct formation of syngas ($H_2$ and CO) in a temperature range between 700 and 900 °C. Steam gasification, presented in Equation (21), also present risks of catalyst structure damage because of high temperature and possible attack Al-O bonds causing catalyst support collapse [160]. Therefore, regeneration via gasification is not suitable for coke removal from zeolite material and no studies dealing with gasification over zeolites have been found in literature. Studies carried out in this field are mostly focused on spent FCC catalysts used in refineries and are consequently out of the scope of this review [161–164].

**Table 5.** Coke gasification reactions using $H_2O$ or $CO_2$.

| | | | |
|---|---|---|---|
| $CO_2$ gasification | $C(s) + CO_2(g) \rightarrow 2CO(g)$ | +172 kJ/mol | (24) |
| Steam gasification | $C(s) + H_2O(g) \rightarrow CO(g) + H_2(g)$ | +131 kJ/mol | (25) |

### 3.3.2. Hydrogenation

Another method used in literature for coke removal over catalytic material is based on the reactivity of coke with hydrogen or light carbonaceous gases such as alkanes [165]. In particular conditions, hydrocracking reactions are observed and coke is decomposed in lighter volatile gases. When hydrogen is used as a reactant, Marecot et al. observed that methane is the only product formed from coke decomposition over $Pt/Al_2O_3$ catalyst [166]. Therefore, the accepted hydrogenation of coke is presented by the following equation:

$$C(s) + 2H_2\ (g) \rightarrow CH_4\ (g). \qquad\qquad -75\ kJ/mol \qquad (26)$$

Nevertheless, Walker et al. demonstrated the low efficiency of this process by comparison with the previously mentioned methods at 800 °C ($O_2$, $H_2O$ and $CO_2$), finding the lowest coke removal performance for $H_2$ treatment [167]. Indeed, to achieve coke removal via hydrogenation, severe temperature or pressure conditions are needed to thwart hydrogen low reactivity. To limit temperature rise due to reaction exothermicity during catalyst regeneration, several works observed that elevated pressure, between 1 and 10 atm, can be applied to achieve coke elimination [168,169]. Moreover, based on the observations of different studies, coke nature and location seem to greatly influence the hydrogenation of coke. In most cases, coke is only partially removed, with an increase of H/C ratio of remaining coke, suggesting that heavy coke partially reacted to form lighter compounds [170]. According to Gnep et al., reactivity of hydrogen with coke compounds over mordenite zeolite is limited to soft coke (high H/C ratio) while heavy polyaromatic molecules remain unreacted [171]. Other studies determined that coke is preferentially removed near Brønsted acid sites on the internal surface of the catalyst while external coke remained unreacted [118,172]. Consequently, complete coke removal via hydrogenation alone is usually not possible unless with severe operating conditions, which could cause catalyst degradation.

### 4. Analytical Techniques

The following section presents the main relevant analytical techniques used in the field of catalyst deactivation and regeneration, from fresh, spent and regenerated catalysts to the analysis of deactivating species. The different possible analytical techniques found in the

literature are categorized by the type of information provided. A particular attention will be given to small-angle X-ray scattering (SAXS), an innovative technique, which may have an interesting application in this field. Use of SAXS has not been applied to the analysis of coked catalyst yet but could appear as an innovative method of interest due to the high precision of structural analysis at a very small scale, which may allow the characterization of both crystalline zeolites and coke molecules structure.

### 4.1. Catalyst Characterization

The following table (Table 6) gathers the main different analytical techniques that are used for the study and characterization of fresh, spent, and regenerated catalysts. A combination of several of these analyses provides a complete description of catalyst characteristics in order to follow the evolution of new virgin catalyst, deactivated catalyst after process use, and recycled catalyst after regeneration.

**Table 6.** Principal analytical techniques used for characterization of fresh and deactivated catalysts.

| Analytical Technique | Related Information |
|---|---|
| X-ray Diffraction (XRD) + SAXS | Crystalline structure: zeolite type, PSD. |
| Electron microscopy (SEM, TEM) | Surface aspect image, particle size. |
| X-ray Fluorescence spectrometry (XRF) | Elemental analysis: Si/Al ratio for zeolites. |
| Physisorption of N2 (or other inert molecule) | Porosimetry analysis: surface area, pore volume |
| Chemisorption of NH3 (or other probe molecule) | Surface acidity: concentration, strength, type. |

### 4.1.1. Structural and Physical Properties

The most commonly used technique for the study of catalyst structure is X-ray Diffraction (XRD), which determines the bulk structure and composition of heterogeneous catalysts with crystalline structures, but also the average crystallite grain size and the particle size distribution (PSD). The characteristic patterns of common zeolite structure are used to identify the catalyst. XRD analysis can be used as a comparative tool between fresh and deactivated catalysts, before and after reaction, to check any possible structural changes or damages. Alvarez et al. observed a structural change of ZSM-5 zeolite from orthorhombic to tetragonal structure using the XRD method [172]. This modification was attributed to zeolite channel occupation by coke molecules formed during the conversion of methanol to hydrocarbon. However, XRD analysis technique is limited to crystalline phases and is not suitable to analyze amorphous or highly dispersed phases. Use of electron microscopy (EM) is also very common as it provides a visual representation of the catalyst surface. Comparison of surface images of the catalyst are often compared to visually illustrate the formation of coke. Thanks to the SEM photographs presented in Figure 18, Lopez et al. observed a difference between fresh (a) and spent (c) surface of ZSM-5 zeolite used for catalytic pyrolysis of plastic wastes [79]. While crystal size of fresh zeolite is in the range 100–300 nm, deposition of coke leads to the agglomeration of particles and to crystal size growth in the range 300–900 nm. Two modes of electron microscopy are possible: by scanning (SEM), used for imaging at a micrometer scale, or by transmitting (TEM), providing images down to the nanometer scale and therefore mostly used for nanosized catalysts such as metal oxide particles, supported metals, and catalysts with nanopores [173,174]. Electron microscopy is often coupled with energy-dispersive analysis of X-rays (EDX) to add elemental data. Catalyst composition provides important data on zeolites such as silica to alumina ratio giving an indication on catalyst acidity [132]. This characteristic is commonly determined with X-ray fluorescence (XRF) spectrometry, which is particularly well suited to investigate the bulk chemical analysis of major elements. XRF is very commonly used as a primary analysis for catalyst characterization. For example, Ajibola et al. used XRF analysis to compare Si/Al ratio of natural and synthetized Y-zeolite Nigerian kaolin zeolite, with 3.22 and 1.45 ratios, furtherly used for catalytic cracking of polyethylene [175].

Many other optical and surface-sensitive techniques are relevant for fresh or deactivated catalyst characterization, which are classified following their capacity to emit or absorb photons (i.e., NMR, IRFT, DRX, UV, etc.), electrons (i.e., XRF, TEM, ESR, etc.) or neutrons and ions (i.e., SIMS, LEIS, etc.) [176,177]. These techniques are used for elemental analysis but can also provide relevant data on the adsorbed species. As an example, Chen et al. used a combination of $^{13}$C NMR, FTIR and UV-Vis to characterize coke formed over a Y-zeolite during catalytic pyrolysis of PE [65]. Deconvolution of NMR spectra provided them with a very precise description of coke nature (aliphatic, aromatic), amount, and repartition.

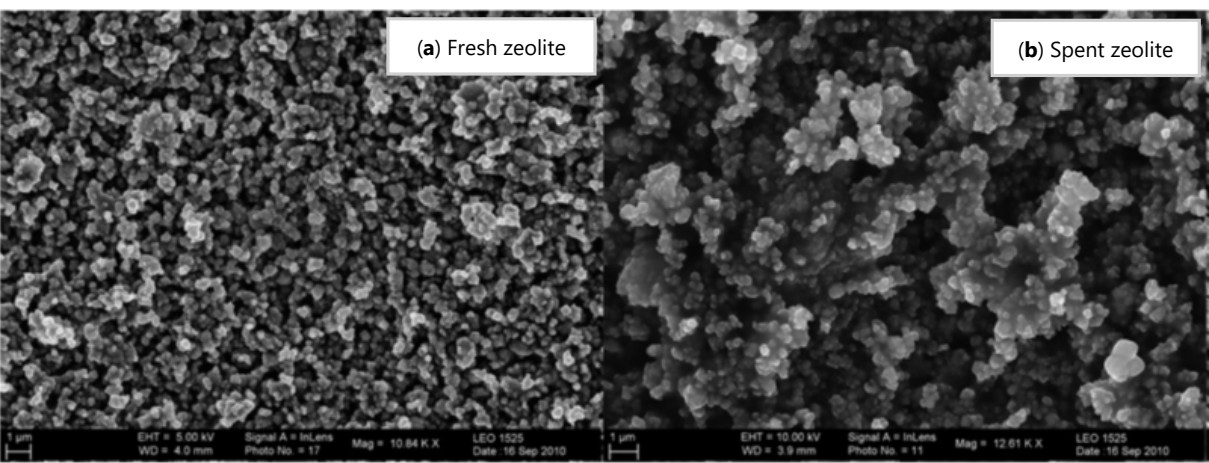

**Figure 18.** SEM images of fresh (**a**) and aged (**b**) ZSM-5 zeolite used for catalytic pyrolysis of plastic wastes. Adapted from Lopez et al. [79].

In spite of the overall structure and composition of the catalyst, the determination of the surface area and volume of pores is very important for the characterization of solid catalysts since these parameters have been proved to be correlated with catalytic activity [77,79]. The most common technique used for this measure is the $N_2$ adsorption and desorption experiment. Comparison of results for a fresh and deactivated catalyst emphasizes the direct impact of carbon deposition on the catalyst surface area and pore characteristics [149]. During the catalytic pyrolysis of PE in a conical-spouted bed reactor carried out by Elordi et al. with different zeolites, coke deposition led to the reduction of the BET surface area, 19% and 39% for HZSM-5 and HY zeolites, respectively, as well as the micropore area [71]. By comparing the results between the fresh and spent catalyst, relative accessibility to catalytic active sites can be determined and coke location can be hypothesized from pore volume repartition and variation. Other inert molecules can be used to get as close as possible to the real reaction conditions: the use of an inert probe molecule having a similar size to the reactant leads to the determination of the surface area effectively accessible during the reaction. For zeolites used for n-heptane cracking, Magnoux et al. used inert n-hexane as a probe having similar dimensions with the reactant and, for some of the investigated zeolites, observed a variation of accessible surface area compared to $N_2$ measurement due to their shape-selectivity [93]. As reported in the following paragraph, adsorption experiments are also useful for the determination of chemical properties of catalysts by analyzing the interactions between the probe molecule and catalytic surface.

### 4.1.2. Chemical Properties and Reactivity

As mentioned previously, using specific probe molecules interacting with active sites in adsorption-desorption experiments ensure the extraction of additional chemical information. This technique, referred to as Temperature-Programmed Desorption (TPD), is commonly used to study the reactivity and acidity of catalytic active sites. The temperature at which desorption occurs indicates the strength of the acid sites while probe molecule

quantity consumed or released is related to their concentration over the external and internal accessible surface of the catalyst. The formation of coke leads to an inevitable decrease of catalytic acid site concentration due to the loss of the active surface. The most common molecules used as probes are $NH_3$ and $CO_2$ for acidic and basic site identification, respectively. Experiments with some other molecules, such as $H_2O$ or pyridine, were also carried out in literature to get more specific information [177]. For instance, using pyridine as probe molecule ensures the differentiation between Brønsted and Lewis acid sites, which is not possible with $NH_3$. Engtrakul et al. used combined $NH_3$-TPD, to quantify the total number of acid sites, and pyridine DRIFTS (diffuse-reflectance FTIR spectroscopy) to probe their nature and repartition (Lewis or Brønsted sites) over a ZSM-5 zeolite used for pine pyrolysis vapor reforming [178]. Moreover, combined reactor and analysis (FTIR, UV-Vis, etc.) over aged catalysts can be used to study the effect of coke over active sites, especially if used operando, offering observation of catalyst acidity evolution and coke nature as deactivation progresses. Goetze et al. used operando UV-Vis spectroscopy to monitor the formation of hydrocarbon pool species leading to the accumulation of coke during the methanol-to-olefins process over HZSM-5 zeolite [179]. Pyridine, being a relatively large molecule, its use is limited to catalysts with particular geometry, especially pore apertures, allowing its diffusion within its pores, and is not applicable to zeolite material with small pores and channels, such as HFER [180]. Consequently, $NH_3$-TPD experiment remains the most commonly used method for catalyst acidity measurement due to its wide application range and to its weak basicity, avoiding coke molecule "replacement" when analyzing aged catalysts. Comparison of fresh and spent catalyst $NH_3$-TPD spectra provide information about the remaining acid sites and therefore catalytic activity. Figure 19 represents comparative TPD experiments for a USHY zeolite deactivated by 1-pentene with and without $NH_3$ probe adsorption. The difference between desorption behavior in the presence or absence of $NH_3$ ensures the determination of free acidity since desorption without a probe molecule is only due to coke [20,181]. From raw TPD experiment curves, deconvolution calculations are necessary to determine the total amount and the strength of catalytic acid sites, as carried out by Khangkham et al. for the characterization of zeolites furtherly used for PMMA cracking [182].

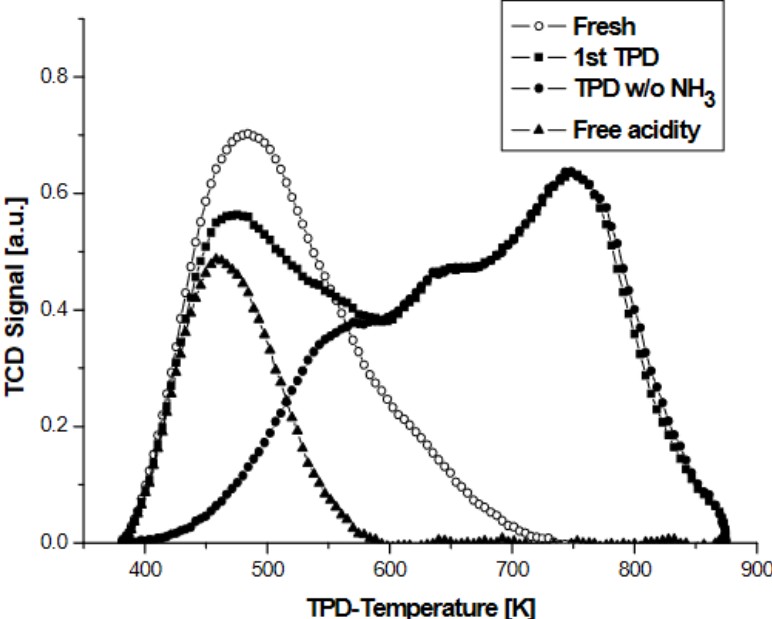

**Figure 19.** Determination of free acidity of deactivated USHY zeolite via TPD experiments (using $NH_3$ as probe molecule). Wang et al. [181].

## 4.2. Coke Analysis

The following table (Table 7) gathers the main different analytical techniques that are used for the study and characterization of coke and deactivating species. Extensive research has been carried out to understand the phenomenon of coke formation, aiming to determine the involved mechanisms and kinetics thanks to one or a combination of these analytical techniques.

**Table 7.** Principal analytical techniques used for the analysis of deactivating species over solid catalysts.

| Analytical Technique | Related Information |
| --- | --- |
| Elemental analysis | C, H, N, S contents<br>H/C ratio |
| Temperature-Programmed Oxidation (TPO) | Global coke content, H/C ratio<br>Coke reactivity and optimal oxidation temperature |
| FTIR, Raman | Coke nature: aliphatic, aromatic<br>Coke effect on active sites |
| UV-Vis | Coke nature: insaturated compounds |
| NMR and XRD | Coke nature and location<br>Structural degradation |
| Coke extraction + analysis (see Figure 22) | Chemical nature of coke<br>Distribution of coke components. |
| Tapered Element Oscillating Microbalance (TEOM) | Coking and deactivation kinetics |
| Thermogravimetric Analysis (TGA) | Coke amount and thermal degradation products |

### 4.2.1. Nature and Composition

Elemental analysis is the first approach to characterize coke in order to obtain a rough composition of species causing deactivation. Commonly used elemental study is carried out by combustion of deactivated catalyst with CHN elemental analysis and provides C, H, N, and S contents. Therefore, this technique determines the H/C ratio, which is the principal characteristic giving indication of the average nature of coke. Elemental analysis is also used to determine regeneration process efficiency via calculating carbon removal proportion. Richard et al. used this criterion to evaluate the influence of ozonation operating parameters for the regeneration of industrial coked zeolites and achieved a maximum of 74.3% carbon removal [149]. The main limitation to this analysis is the potential presence of hydroxyl groups or water molecules within the catalyst that may distort the actual coke composition [183]. Moreover, obtained data is only an average value while coke nature and content may vary alongside the catalyst bed or even within a single pellet because of diffusion limitations.

Temperature-Programmed Oxidation (TPO) analysis provides important data on coke nature showing successive steps during coke combustion [100]. This experiment consists in exposing deactivated catalyst to oxygen flow at varying temperatures. Thanks to mass loss, determined by microgravimetry, and to the formed products during oxidation, commonly analyzed in-line via GC-MS, it is possible to extract information about coke nature. Some data concerning global oxidation mechanisms are also available from TPO curves, typically showing distinct oxidation peaks for hydrogen ($H_2O$ firstly observed) and carbon removal (mixed $CO/CO_2$) [101]. Chen et al. used TPO experiments to characterize coke formed over Y-zeolite during catalytic pyrolysis of PE and observed, from the deconvolution curves (dash lines) presented in Figure 20, two types of coke: external coke with a peak temperature at 400–433 °C and internal coke oxidized at 464–488 °C [65]. TPO analysis is a very widespread method in the field of catalyst deactivation as it provides very complete data concerning coke (global content and H/C ratio) determined from the oxidation products. Even though this technique does not offer precise characterization of coke nature, it is widely used to determine the optimal temperature for regenerative treatments via

oxidation. While oxygen is the most common molecule used, using other molecules is possible and are referred to as Temperature-Programmed Surface Reaction (TPSR) [176].

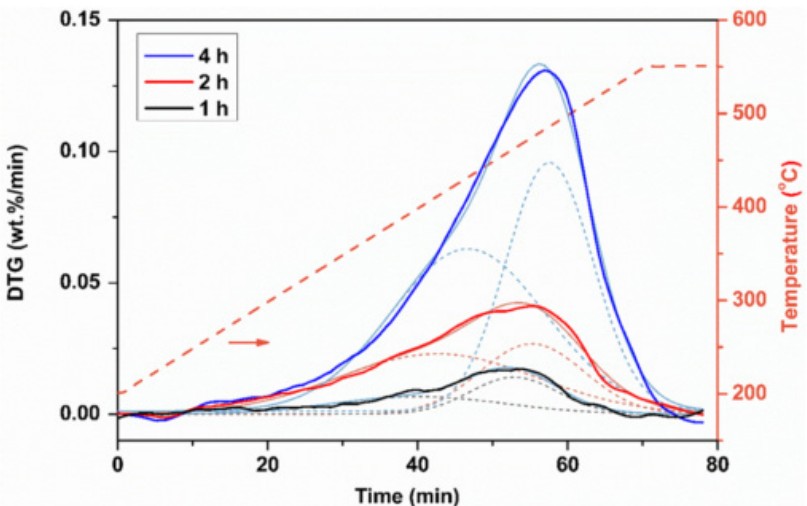

**Figure 20.** TPO curves and deconvoluted peaks of Y-zeolite coked during catalytic pyrolysis of PE at different TOS. Chen et al. [65].

Different analytical techniques are relevant for in situ characterization of coke, for which being non-destructive methods are a main advantage, providing successive analysis for a single sample. IR and UV-Vis spectroscopy, XRD and NMR techniques are commonly used to further investigate coke molecule nature and location. Implementation of operando analysis is very interesting as it becomes possible to simultaneously monitor the reaction advancement and modifications of the catalyst due to coke formation [177]. Guisnet et al. developed a very complete method, presented in Figure 21, to determine the chemical nature and distribution of coke compounds on deactivated catalysts [76]. This method relies on the partial solubility of coke in $CH_2Cl_2$ and on its absence of reactivity with hydrofluoric acid to separate coke from the zeolite structure without modifying the chemical nature of molecules. Soluble coke is afterwards characterized by further analysis such as gas-chromatography coupled with mass spectrometry (GC/MS) or other relevant techniques. However, the heavier fraction of coke may remain insoluble in $CH_2Cl_2$ and analysis is limited to elemental composition and shape location within catalyst pores.

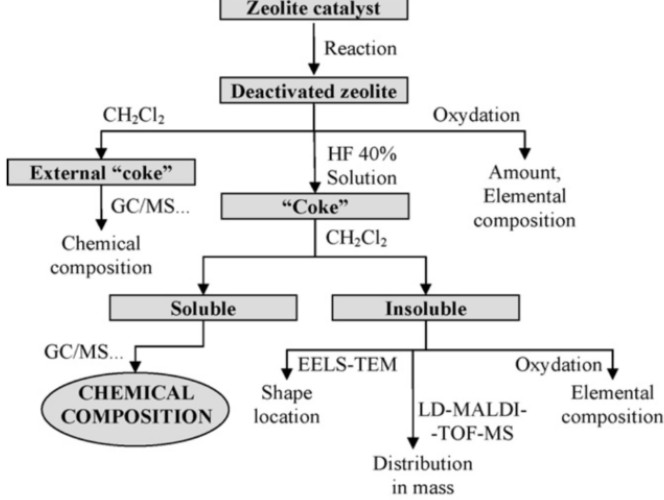

**Figure 21.** Method for the determination of zeolite coke nature and composition developed by Guisnet et al. Graphical abstract from Gusinet et al. [75].

### 4.2.2. Reactivity and Kinetics

Kinetic study of coke formation is challenging due to the complexity of coke formation mechanisms and competing reactions. Therefore, they are not easily determined thanks to analytical methods. The main approach reported in literature is to decompose the overall coke formation into single elementary steps in order to develop precise kinetic models. However, this approach is only a modelling method and is not observable with analytical techniques. Operando mass measurements on microbalance versus time are usually used to determine the overall reaction rate of coke formation. The mass increase of the catalyst during reaction can be related to a reaction rate when reported to a time unit. Conventional microbalance coupled with the reactor are conceivable for in situ monitoring (microgravimetry), but use of Tapered Element Oscillating Microbalance (TEOM) provides more precise measurements thanks to the detection of oscillation frequency variation of the plate caused by sample mass increase [184]. Using this apparatus, Gomm et al. investigated the in situ deactivation of various zeolites during conversion of 2-propanol and were able to correlate the change of reactivity due to coking with mass changes of the catalyst [185].

A similar approach is adopted when using Thermo Gravimetric Analysis (TGA) as a continuous-flow microreactor to monitor coke levels with coupled on-line GC to measure activity [186]. A mathematical model, the Constant-Coke Arrhenius Plot (CCAP), has been modified and used to determine the active site suppression and pore choking during REY zeolite deactivation used in cumene cracking. This method investigates the direct influence of coke formation over catalytic activity. Some experiments transposed this coupled analysis to investigate the evolution of coke levels during catalyst regeneration thanks to a combined TPO-TGA analysis with on-line product analysis [187]. The data obtained by this coupled analysis gave information on the type of carbons obtained at different oxidation temperatures: either amorphous carbons (oxidation at <600 °C) or filamentous carbons (oxidation at >600 °C). Kinetic modelling is often based on TPO spectrum modelling using a linear combination of kinetic power-law expressions and monitoring carbon oxide concentration evolutions ($CO_2$/CO ratio). These models integrate variable input parameters such as oxygen concentration or heating rate influencing oxidation kinetics. Kinetic parameters are determined using models that best fit experimental data [101,106].

### 4.3. Particular Focus: Small Angle X-ray Scattering (SAXS)

Small Angle X-ray Scattering appears as an innovative analytical technique that is relevant to use in the field of catalyst deactivation and regeneration. This analysis describes the morphology of a given material at the scale of 1 to 100 nm and gives much information over its structure, properties being obtained from the evaluation of the measured diffusion profiles [188]. For instance, SAXS technique provides precise descriptions of nanoparticle size, structure, and shape, but also of surface area and pore distribution. This technique is complementary to the Wide Angle X-ray Scattering (WAXS), scanning material structure over a smaller distance (interatomic scale). These techniques are based on the analysis of the elastic scattering of an incident X-ray beam, usually with a wavelength λ = 0.07–0.2 nm, travelling through the analyzed sample. The small angle scattering, between 0.1 and 10° are recorded and treated. While classic microscopy collects diffused beams through a lens to obtain an image, SAXS directly measures the reciprocal space image, as lenses for X-ray recollection do not exist. The experimental set-up, schematically presented in Figure 22, provides the collection of this image by a sensitive detector, which is afterwards computed to represent the diffusion spectrum and SAXS curve.

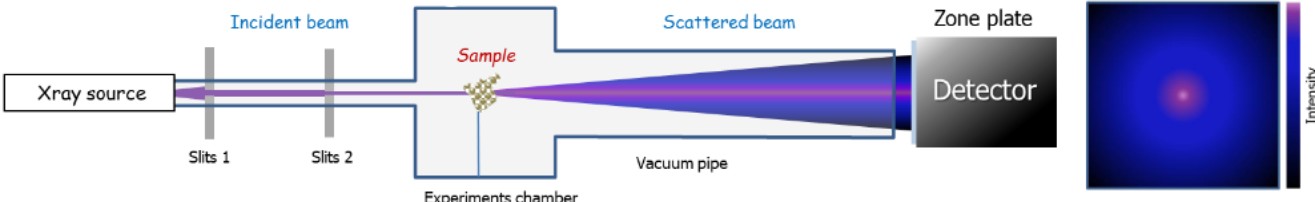

**Figure 22.** Schematic representation of SAXS instrumentation developed by XENOCS.

The obtained signal ensures a description of the material at different scales: as much as the diffusion angle increases, the described distance decreases and becomes more precise. A typical SAXS curve profile is presented in Figure 23, from which three main contributions to the intensity can be observed, represented by the zones with different colors. From intensity singularities within these zones, it is possible to estimate global size of the material, its shape, its surface and finally molecular arrangement of the analyzed material. Different studies intended to develop models describing the different apparent levels of structure, from macropores to atoms [189]. Du et al. used SAXS to precisely characterize microporous zeolites, revealing consistent structural and surface information on the molecular scale [190]. On top of that, in a recent review, Härk et al. emphasized the great efficiency of SAXS for analyzing carbonaceous materials [191]. Saurel et al. successfully applied this technique to ordered and disordered carbonaceous materials and obtained a full description of the pore and atomic structure of the studied compounds [192]. As several studies deal with the separately SAXS analysis of carbonaceous materials and zeolites, SAXS consequently appears as an innovative technique for precise characterization of deactivating coke deposited over zeolite catalysts and is likely to face increasing interest.

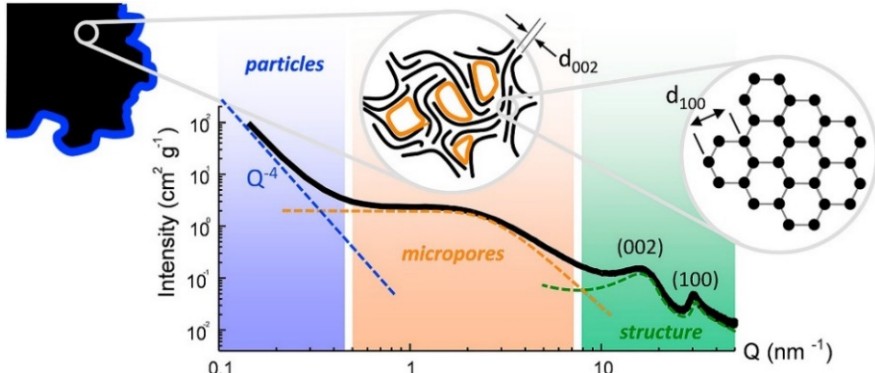

**Figure 23.** Typical SAXS curve representing intensity versus scattering vector, together with schematic example of the structural, microstructural and morphological features of sample determined from SAXS analysis. Saurel et al. [192].

## 5. Conclusions and Perspectives

Recycling of plastic waste has become a major environmental issue over the last decade and the need to develop efficient and viable recycling methods is an actual challenge for handling the growing quantities of accumulated wastes. Catalytic pyrolysis of plastics is a promising process leading to the revalorization of polymer wastes into high-value products with important energetic potential. This review discussed the influence of operating parameters, reactor type and of catalyst use over the repartition and nature of pyrolysis products. Use of zeolite materials appears as an interesting option for catalytic pyrolysis as it yields a high fraction of aromatics. However, the use of this process at an industrial scale is hindered by fast deactivation of catalysts due to coke formation. The complexity of coke formation resides in the diversity of used catalysts (structure and nature), reaction system (reactants, intermediates, and products) and operating parameters (temperature, etc.). The impact of all these parameters over the formation of molecular coke structure have been

widely investigated to determine the influence of coking over catalyst deactivation and loss of activity.

Various methods have been presented in this review for the regeneration and reuse of coked catalysts. Oxidation, gasification, and hydrogenation are the three main processes used in the industry or investigated in research. Even though coke combustion by oxygen remains the most commonly used process in the industry, challenges such as thermal degradation risk or environmental issues led to the need to develop new alternative methods (new processes or combination of existing methods). This review demonstrated that Advanced Oxidation Processes (AOPs) such as ozonation, Non-Thermal Plasma (NTP), or Fenton Process are promising techniques as they allow catalysts regeneration via oxidation under mild conditions. These processes have been previously applied to wastewater and gas treatment but still need further investigation for applications in the field of coke removal from deactivated zeolites, which are used in various processes such as plastic pyrolysis for example. Figure 24 summarizes the main existing methods that have been developed in this review to achieve coke removal.

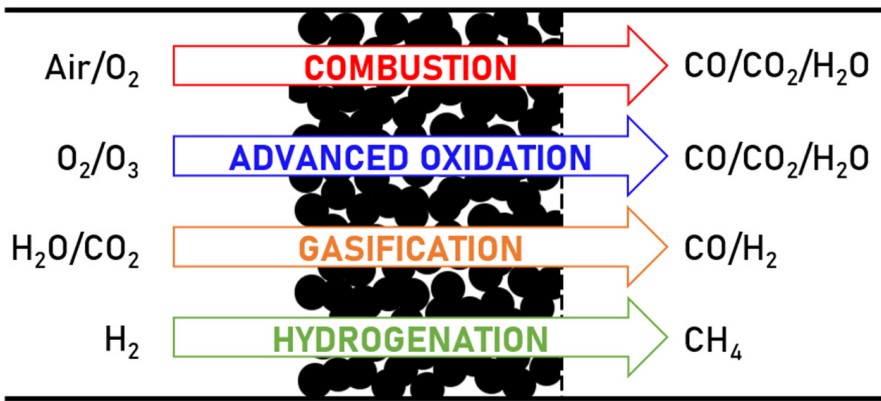

**Figure 24.** Schematic representation of the main coked zeolites regeneration methods with the principal generated products.

Moreover, the relevant analytical techniques in the field of catalyst deactivation and regeneration are summarized to provide a wide landscape of the state of work in the domain and related methodology for past and future studies.

**Author Contributions:** Conceptualization and organization, V.D., R.R. and M.-H.M.; writing— original draft, V.D.; writing—review and editing, V.D., M.-H.M. and R.R. All authors have read and agreed to the published version of the manuscript.

**Funding:** This paper received no external funding.

**Data Availability Statement:** All data were taken from the articles of the bibliography section.

**Conflicts of Interest:** The authors declare no conflict of interest.

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
