# Peer review of "Deactivation and Regeneration of Zeolite Catalysts Used in Pyrolysis of Plastic Wastes—A Process and Analytical Review"

_catalysts, doi:10.3390/catal11070770_

Round 1
Reviewer 1 Report
This review mainly discusses the deactivation and regeneration of zeolite catalysts used in the pyrolysis of plastic wastes. In the beginning, the authors mentioned how important recycling plastic. however, the authors should also add some explanations or examples about the popular catalysts used for this purpose in this decade to show how important the use of Zeolites is. and then start talking about the zeolites.
Besides, for part 2, the authors can consider adjusting their subtitles, part 2.2 may not name the same as part 2.
This manuscript can be considered to be accepted after minor revision.
Author Response
Please find attached the point-by-point response to the reviewer 1's comments

Reviewer 2 Report
This manuscript is a comprehensive review article under the title “Deactivation and regeneration of zeolite catalysts used in pyrolysis of plastic wastes – A process and analytical review”. It is certainly a good effort to summarize the zeolite-based catalysts used in plastic wastes for their deactivation and regeneration phenomenon along with all relevant analytical techniques that can be used to characterize deactivated and regenerated solid catalysts. This manuscript should be accepted after minor revision as the concept and overall writing is beneficial for the scientific community and industry.
Herein, I am summarizing my concerns.
- I guess the figure 1 is taken from somewhere else and the author should acknowledge to the published work by citation such as “idea/figure taken or adopted from the Ref…”
- I have seen some typo errors that should be revised before acceptance such as line 126; nowa days” I would like to suggest for careful revision to make flawless publication.
- The author should consider the connectivity of writing. In all figures taken from other published work should be mentioned with their sources. Such as the figure 12 adapted from Moustafa et al. [83] … but in figure 13 only reference is added with superscript. Keep the same format for all figures.
- The graphical abstract should be revised to understand the general idea of the review to attract more audience.

Author Response
Please find attached the point-by-point response to reveiwer 2's comments
